# Development of an Autonomous Driving Vehicle for Garbage Collection in Residential Areas

**DOI:** 10.3390/s22239094

**Published:** 2022-11-23

**Authors:** Jeong-Won Pyo, Sang-Hyeon Bae, Sung-Hyeon Joo, Mun-Kyu Lee, Arpan Ghosh, Tae-Yong Kuc

**Affiliations:** Department of Electrical and Computer Engineering, College of Information and Communication Engineering, Sungkyunkwan University, Suwon 16419, Republic of Korea

**Keywords:** autonomous control, autonomous vehicles, hardware architectures and software tools, mobility and motion planning, path planning and navigation

## Abstract

Autonomous driving and its real-world implementation have been among the most actively studied topics in the past few years. In recent years, this growth has been accelerated by the development of advanced deep learning-based data processing technologies. Moreover, large automakers manufacture vehicles that can achieve partially or fully autonomous driving for driving on real roads. However, self-driving cars are limited to some areas with multi-lane roads, such as highways, and self-driving cars that drive in urban areas or residential complexes are still in the development stage. Among autonomous vehicles for various purposes, this paper focused on the development of autonomous vehicles for garbage collection in residential areas. Since we set the target environment of the vehicle as a residential complex, there is a difference from the target environment of a general autonomous vehicle. Therefore, in this paper, we defined ODD, including vehicle length, speed, and driving conditions for the development vehicle to drive in a residential area. In addition, to recognize the vehicle’s surroundings and respond to various situations, it is equipped with various sensors and additional devices that can notify the outside of the vehicle’s state or operate it in an emergency. In addition, an autonomous driving system capable of object recognition, lane recognition, route planning, vehicle manipulation, and abnormal situation detection was configured to suit the vehicle hardware and driving environment configured in this way. Finally, by performing autonomous driving in the actual experimental section with the developed vehicle, it was confirmed that the function of autonomous driving in the residential area works appropriately. Moreover, we confirmed that this vehicle would support garbage collection works through the experiment of work efficiency.

## 1. Introduction

The necessity for autonomous driving vehicles is one of the topics that have been constantly raised. Vehicles are mainly used as a means of transportation, and the driver must concentrate on driving while performing movement and transportation. The longer the driving time, the more fatigued the driver, which leads to reduced concentration and poor judgment. So, autonomous driving vehicles are needed to replace human driving and prevent accidents caused by human factors. Another advantage is that the driver no longer has to be tied to the seat, allowing the travel time to be used freely. Moreover, it can significantly improve the mobility of the underprivileged, such as the elderly and the disabled, who have difficulty driving. Unlike humans, additionally, autonomous vehicles on transportation can travel day and night so that even if the driver does not perform arduous long-distance driving, transportation time can be significantly reduced.

Recognizing the environment around the vehicle is essential in composing such an autonomous vehicle. To this end, autonomous vehicles are equipped with sensors such as ultrasound, LiDAR, radar, and cameras. Then, these data go through the process of recognizing the vehicle’s surrounding environment. Recently, with the rapid development of deep learning, the technology for recognizing the surrounding environment through sensor data has also developed significantly. These deep learning technologies are commonly used for recognizing lanes through cameras [1,2,3,4,5,6,7,8,9,10] or for recognizing surrounding objects [11,12,13,14], or as a method for simultaneously recognizing surrounding objects using the camera and lidar [15,16] in autonomous vehicles. Along with various high-precision sensors and deep learning technologies described above, research and commercialization of autonomous vehicles that can autonomously drive on roads without human intervention are also rapidly progressing. Moreover, some global automakers already have highly reliable autonomous driving technologies and are focusing on commercialized vehicles for consumer use.

As of 2020, the top three companies in autonomous driving performance are Waymo, GM, and Zoox, whose autonomous driving distances are approximately 1,300,000, 450,000, and 31,000 miles, respectively, and the number of malfunctions per 1000 miles is 0.09, 0.19, and 0.52, respectively. However, these companies are far from commercial vehicles because they focus on autonomous driving technology and services. Commercialized vehicles are mainly manufactured by professional manufacturers such as Tesla, Volkswagen, and Honda, but they are still driven by human-assisted autonomous driving rather than fully autonomous driving. Moreover, autonomous driving of these vehicles is mainly performed on limited sections of roads such as highways. In addition, drivers must sit in the driver’s seat to cope with unpredictable situations. It means that commercial vehicles’ current autonomous driving system is closer to advanced driver assistance systems (ADAS) than fully autonomous driving.

As mentioned above, autonomous driving vehicles can be applied in various fields. Not only can it be used in ordinary people’s vehicles to perform autonomous driving, but it can also be applied to vehicles or security vehicles to replace human tasks. Primarily, this paper focused on developing an autonomous vehicle to collect garbage, such as household waste, in residential areas. The autonomous driving system in complex environments such as residential areas has to be more dynamic to respond to undue emergencies, as unforeseen events are more likely to occur than on highways. Moreover, for general garbage collection operations, two workers are usually required. One oversees driving, and the other is usually responsible for collecting garbage and loading it into the vehicle. However, suppose the autonomous driving system is adopted. In that case, the workload can be halved as workers do not have to drive the vehicle, so both workers can concentrate on collecting garbage or leave only one worker to collect garbage. Therefore, autonomous vehicles can ideally reduce labor costs and secure economic feasibility in residential areas.

In this paper, we developed an autonomous driving vehicle for garbage collection in residential areas and validated the working of the proposed vehicle in an actual residential area. Before configuring the vehicle, in Section 3, we first defined an operational design domain (ODD) based on our target residential area. In Section 4, we selected and placed the sensors considering the sensor’s field of view (FoV). Furthermore, additional external devices are installed on the vehicle to cope with unexpected situations. Then, in Section 5, we designed an autonomous driving system with vehicle modeling, a recognition algorithm using sensors, a vehicle pose estimation algorithm, a vehicle path planning algorithm, and a safety algorithm. Finally, in Section 6, we experimented with the developed vehicle for position, steering, speed, emergency stop, obstacle recognition, and work efficiency in an actual environment and presented that this development vehicle can be adopted for garbage collection in residential areas.

## 2. Related Works

Various technologies are required to construct autonomous vehicles. Autonomous vehicle driving is possible only when all the essential technologies, such as recognizing the surrounding environment, judging the situation, creating a path, and operating for driving, are combined. Autonomous vehicle driving becomes unstable or impossible if any of these technologies are lacking. Here, we describe related studies on autonomous driving vehicles.

While driving, people usually see lanes and drive along the roads. Likewise, in autonomous driving, lane detection is an essential part of driving along a lane. In the case of lane detection, it is common for autonomous driving systems to recognize lanes through a camera attached to the vehicle, just as a person identifies lanes by looking at them with their eyes. Recently, as research on deep learning has been actively conducted, research on lane detection using it has increased. Neven et al. [1] presented lane recognition by constructing a LaneNet that can immediately obtain the next-best instance segmentation using the clustering loss function. CNN in Yu et al. [6] was constructed by adding a perspective transform layer that enables accurate semantic segmentation of the lane, even when the pixels are reduced according to the distance of the lane seen in the image. Zheng et al. [5] showed a recurrent feature-shift aggregator (RESA) to acquire enriched lane features from general CNN features using the spatial relationships of pixels across rows and columns. Hou et al. [7] applied the self-attention distillation (SAD) method to perform lane detection by improving ENet [2] performance without any additional data or labels.

Other studies have been conducted to solve problems implementing lane detection through networks. Philion [4] developed a semantic segmentation network to increase the accuracy of long-tail lane detection in various environments through synthesized perturbations [17] and domain adaptation using CycleGAN [18]. Liu et al. [10] improved the performance of lane detection using ERFNet [19] and images in low-light conditions generated by the proposed Better-CycleGAN. To overcome the limitations of the camera sensor, Yin et al. [3] presented a lane detection method based on time by applying long short-term memory (LSTM) with the segmentation of the camera and LIDAR bird’s eye view using DeepLabv3+ [20]. Furthermore, Khanm et al. [8] proposed an architecture integrated with VGG16 and gated recurrent unit (GRU) for lane-following on roads. From another perspective, Waykole et al. [9] presented a lane detection algorithm with model predictive control (MPC) for autonomous vehicles on different road pavements to overcome low lane detection accuracy and tracking.

Besides lane detection, understanding the surrounding environment is an essential part of autonomous driving systems. Autonomous vehicles recognize the environment through surrounding sensors, just as humans recognize their surroundings as objects or situations with their eyes immediately. The most used method is semantic segmentation, which recently has achieved considerable improvements with deep learning. To solve the misclassification cases in network-based semantic segmentation using images, Liu et al. [11] presented a severity-aware reinforcement learning using the Wasserstein distance between the semantic segmentation label of the image extracted from the Carla simulator [21]. Chen et al. [12] presented a semantic segmentation method in urban scenes using reality-oriented adaptation networks (ROAD-Net) to convert synthetic data acquired in a virtual environment to an actual data domain. Moreover, Li et al. [14] proposed an Efficient Symmetric Network (ESNet) of a real-time semantic segmentation model for autonomous driving. In addition to images, studies on semantic segmentation using LiDAR have also been conducted. Aksoy et al. [13] developed a semantic segmentation network by constructing an encoder–decoder structured network of 3D LiDAR point clouds, while Cortinhal et al. [15] showed a more robust 3D LiDAR semantic segmentation network composed by applying the context module to the front part of the SalsaNet [13] constructed above. Moreover, some fusion research was also conducted to recognize the surrounding environment. Florea et al. [16] presented a real-time perception component based on the low-level fusion between point clouds and semantic scene information. Furthermore, Hemmati et al. [22] presented an adaptive system with a hardware-software co-design on Zynq UltraScale+ MPSoC that detects pedestrians and vehicles in different lighting conditions on the road.

As described above, current environmental recognition methods, such as lane recognition and semantic segmentation, have achieved many results, according to active research in deep learning. One of the most critical factors in such deep learning networks is the collection of datasets and their labeling. However, it is challenging to construct a dataset because of time, money, and human resources constraints. Therefore, in general, we teach networks to utilize public datasets matching the purposes we want. Recently, many datasets related to autonomous driving have been created and published. Caesar et al. [23] collected data with six cameras, five radars, and one LiDAR every 20 s in 1000 scenes and published a fully annotated dataset with 3D bounding boxes for 23 classes of eight attributes. Instead of the existing simple and small amount lane dataset, Pan et al. [24] published a dataset of 133,235 labeled images for lanes with various shapes extracted from various scenarios and performed lane detection with spatial CNN (SCNN) using a 3-D kernel tensor. Yu et al. [25] collected 100 K driving videos from various weather, scenes, and day hours, which were annotated by scene tagging, object bounding boxes, drivable areas, lane markings, and full-frame instance segmentation.

Another critical element of autonomous driving is determining a vehicle’s current location. When people are driving, they can decide on their location and drive with the help of the GPS and navigation system installed in the vehicle. Similarly, autonomous vehicles determine their location using GPS sensors and HD maps instead of navigation systems. The difference between people and an autonomous vehicle is that people have no hindrance to driving, even if they only know the approximate location. In the case of an autonomous vehicle, it is possible to drive only when the location is precisely determined. However, deciding the accurate location requires a precise GPS sensor, which is very expensive. Therefore, many studies have been conducted to improve the performance by combining a general GPS sensor with a camera sensor. Chen et al. [26] developed a GNSS-Visual-ORB-SLAM (GVORB) method that combines low-cost GNSS and ORB-SLAM [27] for a monocular camera to compensate for the shortcomings of the GNSS sensor. Cai et al. [28] showed the correct localization method of the vehicle using the EKF with the distance between the vehicle location from the GPS sensor and the discrepancies between the lanes in the HD map and those detected by the monocular camera. In addition to camera sensors, many studies have been conducted to improve the accuracy of IMU sensors. Lee et al. [29] improved the localization accuracy through the EKF using the information of the driving lane and stop lane from the lane recognition [30] through a monocular camera to refine errors occurring in GPS, IMU, and onboard odometers. Chu et al. [31] presented the EKF method with a combined monocular camera, IMU, and GNSS to solve problems such as signal attenuation, reflections, or blockages from GNSS and accumulated errors over time from IMU using motion estimation information with SIFT and RANSAC.

After understanding the environment around the vehicle and determining its location of the vehicle, it is necessary to drive the vehicle. While a person can observe the road and make a driving path intuitively, for an autonomous driving system, the driving path must be created for the vehicle to drive. Research on path generation for mobile robots has been actively studied for a long time. Since research on autonomous driving has been active recently, many studies have been conducted to generate a path by applying it to a vehicle. Jinage et al. [32] proposed a hierarchical route-searching algorithm for lane-level route planning for autonomous vehicles and a novel seven-layer map structure called the Tsinghua map model designed from that algorithm. Furthermore, Zhang et al. [33] presented a motion planning method using the C/GMRES algorithm to consider traffic interaction and accelerate calculation. After generating the path, we need to control the autonomous vehicle. Hossain et al. [34] presented a longitudinal and lateral control system for the autonomous vehicle with a hybrid trajectory tracking algorithm that merged with feedforward and feedback control. Moreover, Plessen et al. [35] showed adaptive cruise control (ACC) coupled with obstacle avoidance, which enables autonomous vehicles to drive on regular and curved roads. For this ACC, spatial-based predictive control that applied linear time-varying model predictive control (LTV-MPC) [36] to space rather than time, and geometric corridor planning through graph generation [37] was performed. Furthermore, Li et al. [38] proposed an artificial potential field method with multiple repulsive fields for autonomous trajectory planning and tracking control simultaneously. To control the autonomous vehicle more dynamically, Lee et al. [39] developed a DNN-based nonlinear model predictive control (NMPC) method for the autonomous vehicle to perform a drift maneuver. Besides planning and control, autonomous vehicles are required to drive safely. Aiming to this purpose, Alsuwian et al. [40] presented an advanced emergency braking system (EBS) with multi-sensor fusion to brake the vehicle to avoid or mitigate a collision. Moreover, Diachuk et al. [41] presented a safe and smooth motion planning technique that combined the finite element method (FEM) and nonlinear optimization for autonomous vehicles.

## 3. Operational Design Domain (ODD)

We organized the operational design domain (ODD) for the autonomous vehicle to operate in a residential area comprising a two-lane road moving in each direction. Residential areas were chosen as the desired action area because garbage collection is done mainly from this area. There are specific points within the route for the vehicle to stop and collect the garbage. From Figure 1, it is noticeable that the route consists of two lanes, and the width of the road is also relatively narrow. Therefore, sharp curves can occur frequently. Hence, from the descriptions mentioned above, we can assume that lane changes are impossible, as the lanes are separated by a center-line marking for vehicles to move in each direction. That being the case, using a large vehicle above a length of 5 m on this type of road is also not feasible for the vehicle to navigate autonomously. Therefore, we must limit the length of the autonomous vehicle to less than 4.5 m; so that it can move swiftly.

In a real-world scenario, we can find vehicles parked on the side of the road, further narrowing the path. Henceforth, to avoid such parked vehicles, the center line must be violated; however, as explained earlier, it is a two-lane road, and lane change is not allowed. Therefore, we excluded these scenarios from ODD.

Additionally, we considered that the crosswalks had no traffic lights. Unlike ordinary vehicles, garbage collection vehicles are mostly very slow because workers must check, sort, and load garbage into the vehicle. Therefore, vehicles with autonomous driving must be slower than ordinary vehicles for safety. Therefore, we set the autonomous vehicle’s speed limit to less than 5 km/h in the proposed environment.

Moreover, for the vehicle to safely perform autonomous driving, all vehicles except for the auxiliary test vehicle and the control vehicle were prohibited during the demonstration. Additionally, to maintain safe operation, safety personnel were also deployed at each driving section, such as at the sharp curves, with two additional auxiliary safety personnel present around the vehicle at all times to prevent accidents or safety-related issues.

## 4. Vehicle Hardware

### 4.1. Sensors

As shown in Figure 2, a total of six cameras, four radars, one LiDAR, eight ultrasonic, and one real-time kinematic (RTK) sensor are placed to understand the environment around the vehicle without exception. Before the arrangement, we set and placed the sensors considered with the FoV of the sensors, as shown in Figure 3. Except for one camera (Mobileye, 630 PRO), all other cameras were the same (Leopard Imaging, IMX-390). For the radar, four vehicle-specific radars (continental, SRR-308) were used at each edge of the vehicle to detect objects from all directions. Furthermore, only one LiDAR sensor (Velodyne, VLP-16) was installed, which was placed on the front grille of the vehicle to detect objects and obstacles in front of the vehicle. The ultrasonic sensors (Sensortec, STP-313) were located around the vehicle to detect obstacles close to the vehicle to prevent collisions. We installed the antenna of the RTK module (Synerex, MRP-2000) on the garage collection box to receive the signal at maximum strength.

### 4.2. Module Box

We divided the autonomous driving module into eight sub-modules (Nvidia, AGX Xavier). The reason behind this configuration is that if the camera, LiDAR, radar, ultrasonic, and RTK information are handled in one module, a significant amount of computing power is required, and the price of the host computer also increases rapidly considering the computing power. Therefore, we configured the autonomous driving module with eight sub-modules to efficiently process the information by distributing ample computing power to the sub-modules. As shown in Figure 4, the information of four cameras is distributed by one submodule each. Moreover, other submodules are also separately managed for each process, such as LiDAR, radar, and ultrasonic information. One module was allocated to each camera because the image processing required high computing power than other information. Especially, in this paper, we require more computing power because we implemented the network [42] to detect objects and obstacles in the camera and the network [1] for lane recognition compared to general data processing. In the case of object detection [43] using LiDAR data, it has been confirmed that the computing power of one submodule is sufficient. In addition, radar data are not significant because the sensor module can directly receive information about the object’s information through CAN communication. Moreover, the additional camera (Mobileye, 630 PRO) also does not require significant computing power because it can receive data of lanes and objects directly from the CAN communication. Therefore, a single submodule can simultaneously manage the additional camera and fallback processing. The other submodule is allocated to drive the central driving system and simultaneously receives sensor data from the ultrasonic sensor through LIN communication. We use Ethernet for communication between the submodules, and the command values required to drive the vehicle are directed to the controller through CAN communication.

### 4.3. User Interface

The user interface (UI) for our autonomous vehicle is aimed at monitoring the status of the autonomous driving system and operating the driving system to implement the vehicle accurately. The left panel in Figure 5 displays the status of the vehicle’s connection between the controllers, sensors, and autonomous driving system. The LED indicator of each item is green when it is working correctly and red when it is in an abnormal state, such as disconnected or failing to obtain any sensor data. If at least one of the LED indicators is abnormal, it sends the vehicle a stationary emergency signal to the autonomous driving system. In contrast, the middle panel displays a high-definition (HD) map, as shown in Figure 5. The middle panel also displays the location of the vehicle, planned route, and crosswalks on the map. To create a planned path, we determine the target location on the HD map by clicking on the middle panel. In the case of the right panel in Figure 5, buttons are used to control the driving mode and direction indicators of the vehicle at the top, and the lower part is configured to check the camera feed installed on the vehicle. The operator can check the sensor connection status or controller status abnormality by looking at this UI and altering the vehicle’s mode by changing the vehicle’s status into manual mode from autonomous driving mode.

### 4.4. Control Button

Safety should be the top priority of autonomous vehicles. Although the probability of a safety accident is low because the vehicle in this paper has a shallow speed, there is still the possibility of an accident. Therefore, we installed additional control buttons on the vehicle, as shown in Figure 2. These control buttons can operate the vehicle from the inside, the outside, and the remote controller. In the case of an emergency, safety personnel can prevent safety-related incidents in advance by pressing any of one additionally equipped buttons.

In this paper, an autonomous vehicle has two modes. One is the manual mode, and the other is the autonomous driving mode. In this vehicle, the autonomous driving mode should be configured by a person from the driver’s seat or the passenger’s seat through an autonomous driving UI. However, even if the autonomous driving mode is set, the vehicle does not start to work immediately because the worker may need some time to get off the vehicle and return to his earlier position. Therefore, we added a run/stop button that needs to be pressed to resume the vehicle’s autonomous driving, hence giving workers complete flexibility to work.

In an autonomous vehicle, if any dangerous situation is detected, the autonomous driving system automatically avoids or stops the vehicle from a risk factor as it can recognize the surrounding environment and detect any dangerous or unusual circumstances. However, regardless of how well the self-driving vehicle performs, there is always the possibility of a system misjudgment or malfunction occurring. No vehicle system is impeccable and can guarantee that the autonomous driving system will always make accurate judgments and perform entirely, even if the environment is fully recognized. Therefore, we installed a fail-safe mechanism and emergency stop buttons in this vehicle. When the emergency stop button is pressed, the vehicle immediately performs an emergency stop. This button operates in both manual and autonomous driving modes. The only way to clear this emergency state is through the autonomous driving UI inside the vehicle.

Additionally, we installed an emergency parking brake button to counter specific emergencies in which the vehicle loses control or functionality. This button is physically connected to the parking brake signal line of the vehicle, and the parking brake is operated immediately when the button is pressed.

### 4.5. Beacon with Sound

The most significant notification factor for the outside workers and safety staff around the vehicle is to recognize which mode the vehicle currently possesses. To check this, we installed beacons to caution the operators so they can more intuitively perceive these conditions rather than constantly surveying the status indications on the screen. From Figure 2, we can see that the beacon is composed of two pieces of green and orange light and is placed on the top of the vehicle so that it can be seen clearly from a healthy distance. If the vehicle’s mode is manual, the green beacon comes off; if the mode is autonomous driving, the green beacon comes on. Furthermore, if the vehicle is in a normal state, the orange beacon stays off, and if the state is in an emergency stop, the orange beacon turns on. However, it is not always possible for the operators to stare at the beacon at all times, so it is difficult to identify immediately if an emergency stop occurs when not looking at the beacon. Therefore, in such an emergency stop state, we also made an audible warning sound so that the operator and the safety staff outside the vehicle could check the vehicle’s current state more intuitively and reliably.

## 5. Vehicle Software

### 5.1. Overall System

We composed our autonomous driving system as shown in Figure 6. An autonomous driving system perceives the surrounding environment using various sensor data. The perception module recognizes lane information, object information, and roads through the segmentation data of the surrounding environment. Then, the recognized information is used in the pose estimator and behavior planner modules. The pose estimation module continuously estimates the vehicle’s pose through various sensor data integrated with the vehicle speed steering status, HD map information, and the information recognized by the perception module. From the HD map, a global path can be generated in the global path planner module based on the estimated vehicle pose, and this path can be transferred to the behavior planner module.

In the behavior planner module, the vehicle’s behavior is generated through the transmitted global path, map information, and recognition information. Based on this generated behavior, the behavior coordinator module creates coordinates of waypoints for the vehicle to proceed. The created coordinates are also transmitted to the vehicle by creating a speed and a steering profile for moving the vehicle through the behavior executor module. Similarly, the emergency behavior planner module receives the sensor information, recognition information, and vehicle pose estimation information to determine and plan actions for emergencies to prevent accidents. Here, emergencies are determined from a distance between obstacles and the vehicle based on the vehicle’s pose, an abnormality status from the sensors, or a signal of control buttons.

### 5.2. Perception Module

As mentioned earlier, knowing the surrounding environment for autonomous vehicle driving is crucial. Among the surrounding environment, two factors that significantly affect autonomous driving are object recognition and lane recognition. As shown in Figure 7, the recognition module in this paper performs object recognition and lane recognition based on camera, radar, and LiDAR data. After this, the information processed in the recognition module is used for pose estimation and vehicle behavior planning.

#### 5.2.1. Object Detection

The Figure 7 shows that camera, radar, and lidar data are all used for object recognition. Here, we use camera and radar data together and lidar data alone. The reason for this is that distance information cannot be accurately known only by the camera recognition result. As a result of camera recognition, the pixel position of an object can be estimated. However, an additional method, such as a stereo camera or a depth estimation algorithm, is required to know the accurate distance information. To overcome this, we identified the object’s position that can be obtained through camera recognition through the distance and angle information of the cluster obtained from the radar data. Second, the camera and radar data recognition results are relatively unstable than the lidar data. In the case of camera recognition, misrecognition always exists because a network-based method is used. In the case of radar data, the noise is severe and causes more noisy data because the sensor performs the clustering. On the other hand, since lidar uses light reflection, it has fewer errors than the above case to perform object recognition more reliably. Therefore, in this paper, object recognition using only LiDAR data was performed separately to detect surrounding objects more stably.

In the case of object recognition through camera and radar data, the object is recognized relatively simply. First, cameras recognize objects using YOLO [42]; the bounding box based on pixel coordinates and class information can be obtained accordingly. On the other hand, in the case of radar, clustering is performed to provide information on the range, azimuth, and relative velocity for each cluster based on a two-dimensional coordinate. Because the cluster information provided in this way is expressed as a single point, it appears as an entire column after being projected in the image plane. Eventually, each bounding box’s distance is given by the most overlapped projected column. In the case of object recognition through lidar, the object was recognized in the same way as [44]. Here, we performed the clustering using DBSCAN [45] instead through SVM after the two-dimensional projection in the previous paper because we aimed at the outdoor environment, while the previous paper performed at an indoor environment.

#### 5.2.2. Lane Detection

We used a combination of three types of sensor information to recognize the lanes. First, lane information is recognized from a binary image. Second, the lane information is obtained through a lane detection network [1], and lastly, the lane information is obtained from a lane recognition camera. To recognize lanes in a binary image, we used a method [46] of extracting polynomial lines using a sliding window technique after applying a perspective transformation to the binary image obtained from the front camera, as shown in Figure 8. However, in this method, recognition failure cases occur because of the light intensity and the lane marking condition. If the lane has a sharp curve, recognition is also affected. To counter these visual shortcomings, we adjusted a more robust lane detection method using a neural network from Neven et al. [1]. However, although this method is less affected by various lighting conditions or lane marking than the lane recognition method using the binary image, it is still not perfect. In the case of a sharp curve, this method has a similar problem to the previous method. Additionally, the lane recognition camera used cannot recognize lanes ideally. Therefore, like the above two methods, it is greatly influenced by lighting conditions or lane markings. Thus, to perform lane recognition better in each frame, we used all three pieces of information together to recognize the lanes as much as possible.

This paper estimates all lanes as second-order polynomial equations for each frame. All our lanes are drawn using the following equation: $x=ay^{2}+by+c$. Here, we use three types of information to recognize lanes, as mentioned above. We recognize the closest left and right lane on each piece of information. We define the three types of information:(1)$$L_{b}=\left[\begin{array}{ccc} a_{b_{r}} & b_{b_{r}} & c_{b_{r}}\\ a_{b_{l}} & b_{b_{l}} & c_{b_{l}} \end{array}\right]$$
(2)$$L_{n}=\left[\begin{array}{ccc} a_{n_{r}} & b_{n_{r}} & c_{n_{r}}\\ a_{n_{l}} & b_{n_{l}} & c_{n_{l}} \end{array}\right]$$
(3)$$L_{m}=\left[\begin{array}{ccc} a_{m_{r}} & b_{m_{r}} & c_{m_{r}}\\ a_{m_{l}} & b_{m_{l}} & c_{m_{l}} \end{array}\right]$$
where *b*, *n*, *m*, *l*, and *r* are the binary information, network information, lane camera information, left lane, and right lane, respectively. As mentioned above, lane recognition could be better. Even if lane recognition is found in the lanes ideally in the t−1 frame, we cannot ensure that lane recognition can find the lanes in *t* frames. To address this problem, we used the queuing method. We store the lane recognition result in the queue after comparing the lane information already stored in the queue and the lane information recognized in this frame. To compare with the lane information in the current frame, we create a new lane from the queue.
(4)$$L_{q_{l}}=\sum_{i=q}^{size(Q_{l})}Q_{l}\left(i\right)/q$$
(5)$$L_{q_{r}}=\sum_{i=q}^{size(Q_{r})}Q_{r}\left(i\right)/q$$
where $L_{q_{l}}$ and $L_{q_{r}}$ are the new lanes from the left and right lanes, respectively. We use the queues separately with the left and right lanes because the lanes sometimes have different shapes. As shown in Equations (Equation 4) and (Equation 5), the new lanes from queues are composed of a few recent results concerning a size of *q*, where the user-specific parameter. This user-specific parameter *q* depends on the processing time of the lane-recognition process.

We also need to compare whether adding queues for lane recognition is worth it. Then, we use an intersection over union (IoU) method to compare the new lanes and the lane recognition results. We assume that our lane recognition results are composed of the image coordinates. Accordingly, we cannot directly compare these two lanes using the IoU method. Hence, we adjust the margin value m around the lanes to create their regions, as shown in Figure 9. When we adjust the margin as above, we can also create boundary lines around this margin as:(6)$$L_{q_{l}}^{\pm }=a_{q_{l}}y^{2}+b_{q_{l}}y+c_{q_{l}}\pm m$$
(7)$$L_{q_{r}}^{\pm }=a_{q_{r}}y^{2}+b_{q_{r}}y+c_{q_{r}}\pm m$$
(8)$$L_{d_{l}}^{\pm }=a_{d_{l}}y^{2}+b_{d_{l}}y+c_{d_{l}}\pm m$$
(9)$$L_{d_{r}}^{\pm }=a_{d_{r}}y^{2}+b_{d_{r}}y+c_{d_{r}}\pm m$$
where *d* is the lane from the detection results. As shown in Equations (Equation 6)–(Equation 9), we can make two boundary lines that divide the plus and minus lines from each lane with the margin value *m*.

As we already know, we can obtain the IoU value by dividing the intersection space by the total space. Now, our lanes have their own spaces, and we can adapt these to obtain the IoU value. From the boundary lines of the margin spaces, we derive each space through an integral as follows:(10)$$\begin{array}{cc} S_{q_{l}} & =\int_{0}^{h}\left(L_{q_{l}}^{+}-L_{q_{l}}^{-}\right)dy \end{array}$$
(11)$$\begin{array}{cc} S_{q_{r}} & =\int_{0}^{h}\left(L_{q_{r}}^{+}-L_{q_{r}}^{-}\right)dy \end{array}$$
(12)$$\begin{array}{cc} S_{d_{l}} & =\int_{0}^{h}\left(L_{d_{l}}^{+}-L_{d_{l}}^{-}\right)dy \end{array}$$
(13)$$\begin{array}{cc} S_{d_{r}} & =\int_{0}^{h}\left(L_{d_{r}}^{+}-L_{d_{r}}^{-}\right)dy \end{array}$$
where *h* denotes the height of the image. Because all our lanes are projected onto an image coordinate, our necessary process is based on the image space.

Now, we consider how to obtain the intersection space between these two spaces. As shown in Figure 9, the green space represents the intersection space. From there, we focus on margin lines. In the outer space of the intersection space, the intersection space follows a line with a smaller *x* value than the other line. On the other hand, in the inner space of the intersection space, the intersection space follows a line with a more significant *x* value than the other line. Therefore, we derive an equation from obtaining the intersection space by considering the above concept.
(14)$$\begin{array}{cc} I_{{\left(q,d\right)}_{l}}= & \int_{0}^{h}\left(min\left(L_{q_{l}}^{+},\;L_{d_{l}}^{+}\right)-max\left(L_{q_{l}}^{-},\;L_{d_{l}}^{-}\right)\right)dy \end{array}$$
(15)$$\begin{array}{cc} I_{{\left(q,d\right)}_{r}}= & \int_{0}^{h}\left(min\left(L_{q_{r}}^{+},\;L_{d_{r}}^{+}\right)-max\left(L_{q_{r}}^{-},\;L_{d_{r}}^{-}\right)\right)dy \end{array}$$

Then, our IoU can be expressed as:(16)$$IoU_{{\left(q,d\right)}_{l}}=\frac{I_{{\left(q,d\right)}_{l}}}{\left(S_{q_{l}}+S_{d_{l}}\right)-I_{{\left(q,d\right)}_{l}}}$$
(17)$$IoU_{{\left(q,d\right)}_{r}}=\frac{I_{{\left(q,d\right)}_{r}}}{\left(S_{q_{r}}+S_{d_{r}}\right)-I_{{\left(q,d\right)}_{r}}}$$

In our case, we use three-lane information instead of using only one-lane information. Here, we compared the three types of information with the lane from the queue because we needed to determine whether the lane information was adequately recognized. If the IoU value from these comparisons exceeds a threshold, then the lane information can be added to the queue. However, we cannot only pick one-lane information according to the IoU values because a higher IoU value is not a guarantee for an actual lane within all over-threshold lanes. Therefore, we aimed to derive lane information by simultaneously adding this information to the queue.
(18)$$L_{d_{l}}=\left\{\begin{array}{cc} L_{b_{l}}, & IoU_{{\left(q,b\right)}_{l}}>s\\ L_{n_{l}}, & IoU_{{\left(q,n\right)}_{l}}>s\\ L_{m_{l}}, & IoU_{{\left(q,m\right)}_{l}}>s\\ \frac{L_{b_{l}}+L_{n_{l}}}{2}, & IoU_{{\left(q,b\right)}_{l}},IoU_{{\left(q,n\right)}_{l}}>s\\ \frac{L_{b_{l}}+L_{m_{l}}}{2}, & IoU_{{\left(q,b\right)}_{l}},IoU_{{\left(q,m\right)}_{l}}>s\\ \frac{L_{n_{l}}+L_{m_{l}}}{2}, & IoU_{{\left(q,n\right)}_{l}},IoU_{{\left(q,m\right)}_{l}}>s\\ \frac{L_{b_{l}}+L_{n_{l}}+L_{m_{l}}}{3}, & \begin{array}{cc} \; & IoU_{{\left(q,b\right)}_{l}},IoU_{{\left(q,n\right)}_{l}},\\ \; & IoU_{{\left(q,m\right)}_{l}} \end{array}>s \end{array}\right.$$
(19)$$L_{d_{r}}=\left\{\begin{array}{cc} L_{b_{r}}, & IoU_{{\left(q,b\right)}_{r}}>s\\ L_{n_{r}}, & IoU_{{\left(q,n\right)}_{r}}>s\\ L_{m_{r}}, & IoU_{{\left(q,m\right)}_{r}}>s\\ \frac{L_{b_{r}}+L_{n_{r}}}{2}, & IoU_{{\left(q,b\right)}_{r}},IoU_{{\left(q,n\right)}_{r}}>s\\ \frac{L_{b_{r}}+L_{m_{r}}}{2}, & IoU_{{\left(q,b\right)}_{r}},IoU_{{\left(q,m\right)}_{r}}>s\\ \frac{L_{n_{r}}+L_{m_{r}}}{2}, & IoU_{{\left(q,n\right)}_{r}},IoU_{{\left(q,m\right)}_{r}}>s\\ \frac{L_{b_{r}}+L_{n_{r}}+L_{m_{r}}}{3}, & \begin{array}{cc} \; & IoU_{{\left(q,b\right)}_{r}},IoU_{{\left(q,n\right)}_{r}},\\ \; & IoU_{{\left(q,m\right)}_{r}} \end{array}>s \end{array}\right.$$
where $L_{b}$, $L_{n}$, $L_{m}$, and *s* are the binary lane, network lane, lane from the camera, and the predefined threshold for IoU, respectively. All the above definitions are possible only when at least one of the IoUs of the comparison between the queue and lane information exists. However, there are cases in which all recognition fails during actual driving. Moreover, even if the lane is recognized well, the lane cannot be added to the queue if the lane that is already stored in the queue is incorrect or all IoU values are below the threshold. Thus, selecting a lane to be added to the queue is a challenging task. To cope with the above case, we temporarily created another queue. This queue stores $L_{d}$ in a temporary queue, even if $L_{d}$ cannot be added to the existing queue. In this case, $L_{d}$ is:(20)$$L_{d_{l}}=\left\{\begin{array}{cc} \frac{L_{b_{l}}+L_{n_{l}}}{2}, & IoU_{{\left(b,n\right)}_{l}}>s\\ \frac{L_{n_{l}}+L_{m_{l}}}{2}, & IoU_{{\left(n,m\right)}_{l}}>s\\ \frac{L_{b_{l}}+L_{m_{l}}}{2}, & IoU_{{\left(b,m\right)}_{l}}>s\\ \frac{L_{b_{l}}+L_{n_{l}}+L_{m_{l}}}{3}, & \begin{array}{cc} \; & IoU_{{\left(b,n\right)}_{l}},IoU_{{\left(b,m\right)}_{l}}\;or\\ \; & IoU_{{\left(b,n\right)}_{l}},IoU_{{\left(n,m\right)}_{l}}\;or\\ \; & IoU_{{\left(b,m\right)}_{l}},IoU_{{\left(n,m\right)}_{l}} \end{array}>s\\ \mathrm{default}\;\mathrm{line}, & \begin{array}{cc} \; & IoU_{{\left(q,b\right)}_{l}},IoU_{{\left(q,n\right)}_{l}}\\ \; & IoU_{{\left(q,m\right)}_{l}} \end{array}<s \end{array}\right.$$
(21)$$L_{d_{r}}=\left\{\begin{array}{cc} \frac{L_{b_{r}}+L_{n_{r}}}{2}, & IoU_{{\left(b,n\right)}_{r}}>s\\ \frac{L_{n_{r}}+L_{m_{r}}}{2}, & IoU_{{\left(n,m\right)}_{r}}>s\\ \frac{L_{b_{r}}+L_{m_{r}}}{2}, & IoU_{{\left(b,m\right)}_{r}}>s\\ \frac{L_{b_{r}}+L_{n_{r}}+L_{m_{r}}}{3}, & \begin{array}{cc} \; & IoU_{{\left(b,n\right)}_{r}},IoU_{{\left(b,m\right)}_{r}}\;or\\ \; & IoU_{{\left(b,n\right)}_{r}},IoU_{{\left(n,m\right)}_{r}}\;or\\ \; & IoU_{{\left(b,m\right)}_{r}},IoU_{{\left(n,m\right)}_{r}} \end{array}>s\\ \mathrm{default}\;\mathrm{line}, & \begin{array}{cc} \; & IoU_{{\left(q,b\right)}_{r}},IoU_{{\left(q,n\right)}_{r}}\\ \; & IoU_{{\left(q,m\right)}_{r}} \end{array}<s \end{array}\right.$$
where the default line is a straight line related to the width of the public road (a=0, b=0, and c=± (roadwidth)/2). If the main queue is utterly different from the lane recognition results during a few frames, it is assumed that the actual lanes are the same as the lane recognition results. Therefore, we replace the main queue as a temporary queue if the temporary queue is larger than the threshold *t*. Subsequently, the temporary queue is initialized.

### 5.3. Vehicle Pose Estimation (VPE)

In the autonomous driving system, it is crucial to know the vehicle’s location on the map. In this paper, we estimated the vehicle’s location using the RTK module. However, some problems arise when the vehicle’s location is determined only through the RTK module. First, the information from the RTK module may not be received because of connection errors or other conditions. If the vehicle only relies on the RTK module, it quickly loses its position as soon as the location value is not received from the RTK module for any reason. In this situation, the vehicle cannot perform autonomous driving and must be stopped. Second, the positional accuracy of the vehicle is determined entirely by the RTK module. Although the location value is received from the RTK module, if the wrong location value is received or the precision of the received location value is low, the vehicle’s position is immediately shifted. At this time, the vehicle is also not in a state that can perform autonomous driving, so the vehicle must be stopped.

To solve this problem, we propose a vehicle pose estimation (VPE) system composed of an extended Kalman filter (EKF). As shown in Figure 10, the EKF in this paper comprises three pieces of information. The first uses the vehicle’s kinematics model information, the second uses the location information obtained from the RTK module, and finally, the third uses the lane information obtained from the camera. The vehicle’s kinematics model, along with the vehicle’s speed and steering angle, is used to estimate the vehicle’s pose through dead reckoning. In addition, it estimates and updates the vehicle’s pose through periodically obtainable location information from the RTK module and lane recognition information.
(22)$$\begin{array}{cc} {\vec{x}}_{k} & ={\left(x_{k}\;y_{k}\;\theta_{k}\;{\dot{x}}_{k}\;{\dot{y}}_{k}\;{\dot{\theta }}_{k}\right)}^{t} \end{array}$$
(23)$$\begin{array}{cc} u_{k} & ={\left(v_{k}\;\delta_{k}\right)}^{t} \end{array}$$
(24)$$\begin{array}{cc} {\hat{\vec{x}}}_{k}^{\;\bar{}} & =f({\hat{\vec{x}}}_{k-1},u_{k}) \end{array}$$
(25)$$\begin{array}{cc} P_{k}^{\;\bar{}} & =A_{k}P_{k-1}A_{k}^{T}+Q_{k} \end{array}$$
(26)$$\begin{array}{cc} K_{k} & =P_{k}^{\;\bar{}}H_{k}^{T}{\left(H_{k}P_{k}^{\;\bar{}}H_{k}^{T}+R_{k}\right)}^{-1} \end{array}$$
(27)$$\begin{array}{cc} {\hat{\vec{x}}}_{k} & ={\hat{\vec{x}}}_{k}^{\;\bar{}}+K_{k}\left(z_{k}-h\left({\hat{\vec{x}}}_{k}^{\;\bar{}}\right)\right) \end{array}$$
(28)$$\begin{array}{cc} P_{k} & =P_{k}^{\;\bar{}}-K_{k}H_{k}P_{k}^{\;\bar{}} \end{array}$$

As mentioned above, we estimated the vehicle’s pose through the EKF. In this paper, we define the state vector ${\vec{x}}_{k}$ and control vector $u_{k}$ using Equations (Equation 22) and (Equation 23), respectively. Here, ${\vec{x}}_{k}$ represents the state of the vehicle to be estimated at time *k*, and $u_{k}$ represents the speed and rotation of the vehicle at time *k*. As shown in Equations (Equation 24) to (Equation 28), we predict ${\vec{x}}_{k}$ through vehicle kinematics in Figure 11, estimate ${\vec{x}}_{k}$ using the Kalman gain *K* and measurement vector $z_{k}$, and update the process covariance matrix *P*.
(29)$$\begin{array}{cc} {\hat{\vec{x}}}_{k}^{\;\bar{}} & =f({\hat{\vec{x}}}_{k-1},u_{k})\\ \left[\begin{array}{c} {\hat{x}}_{k}^{\;\bar{}}\\ {\hat{y}}_{k}^{\;\bar{}}\\ {\hat{\theta }}_{k}^{\;\bar{}}\\ {\hat{\dot{x}}}_{k}^{\;\bar{}}\\ {\hat{\dot{y}}}_{k}^{\;\bar{}}\\ {\hat{\dot{\theta }}}_{k}^{\;\bar{}} \end{array}\right] & =\left[\begin{array}{c} {\hat{x}}_{k-1}\\ {\hat{y}}_{k-1}\\ {\hat{\theta }}_{k-1}\\ {\hat{\dot{x}}}_{k-1}^{\;\bar{}}\\ {\hat{\dot{y}}}_{k-1}^{\;\bar{}}\\ {\hat{\dot{\theta }}}_{k-1}^{\;\bar{}} \end{array}\right]+\left[\begin{array}{c} v_{k}\cos\left({\hat{\theta }}_{k-1}+\delta_{k}\right)\cdot \Delta t\\ v_{k}\sin\left({\hat{\theta }}_{k-1}+\delta_{k}\right)\cdot \Delta t\\ v_{k}\sin(\delta_{k})L\cdot \Delta t\\ v_{k}\cos\left({\hat{\theta }}_{k-1}+\delta_{k}\right)\\ v_{k}\sin\left({\hat{\theta }}_{k-1}+\delta_{k}\right)\\ v_{k}\sin(\delta_{k})L \end{array}\right] \end{array}$$
(30)$$\begin{array}{cc} A_{k} & =\frac{\partial f}{\partial x}{|}_{x={\hat{\vec{x}}}_{k-1}}\\ \; & =\left[\begin{array}{cccccc} 1 & 0 & -v_{k}\sin\left({\hat{\theta }}_{k-1}+\delta_{k}\right)\cdot \Delta t & 0 & 0 & 0\\ 0 & 1 & v_{k}\cos\left({\hat{\theta }}_{k-1}+\delta_{k}\right)\cdot \Delta t & 0 & 0 & 0\\ 0 & 0 & 1 & 0 & 0 & 0\\ 0 & 0 & -v_{k}\sin\left({\hat{\theta }}_{k-1}+\delta_{k}\right) & 1 & 0 & 0\\ 0 & 0 & v_{k}\cos\left({\hat{\theta }}_{k-1}+\delta_{k}\right) & 0 & 1 & 0\\ 0 & 0 & 0 & 0 & 0 & 1 \end{array}\right] \end{array}$$

In Equation (Equation 29), we used vehicle kinematics as shown in Figure (Figure 11) to obtain the predicted state vector ${\hat{\vec{x}}}_{k}^{\;\bar{}}$. Because we used the EKF for non-linear estimation in our work, we calculated the state transition matrix $A_{k}$ by expressing it as a Jacobian matrix of ${\hat{\vec{x}}}_{k-1}$ for the function *f*, as in Equation (Equation 30).
(31)$$\begin{array}{cc} z_{k} & =h({\hat{\vec{x}}}_{k}^{\;\bar{}})\\ \left[\begin{array}{c} x_{rtk}\\ y_{rtk}\\ \theta_{diff} \end{array}\right] & =\left[\begin{array}{c} {\hat{x}}_{k}^{\;\bar{}}-L_{f}\cos{\hat{\theta }}_{k}^{\;\bar{}}\\ {\hat{y}}_{k}^{\;\bar{}}-L_{f}\sin{\hat{\theta }}_{k}^{\;\bar{}}\\ {\hat{\theta }}_{k}^{\;\bar{}}-\theta_{l} \end{array}\right] \end{array}$$
(32)$$\begin{array}{cc} H_{k} & =\frac{\partial h}{\partial x}{|}_{x={\hat{\vec{x}}}_{k}^{\;\bar{}}}\\ \; & =\left[\begin{array}{cccccc} 1 & 0 & L_{f}\sin{\hat{\theta }}_{k}^{\;\bar{}} & 0 & 0 & 0\\ 0 & 1 & -L_{f}\cos{\hat{\theta }}_{k}^{\;\bar{}} & 0 & 0 & 0\\ 0 & 0 & 1 & 0 & 0 & 0 \end{array}\right] \end{array}$$

The measurement vector $z_{k}$ used in the VPE in this paper is given by Equation (Equation 31). All parameters in this equation are based on the center of the front wheel axle of the vehicle. However, the *x* and *y* values obtained from the RTK module are based on the location of the GPS antenna. As shown in Figure 12a, the GPS antenna is located at a specific distance $L_{f}$ behind the center of the front wheel axis to the vehicle center axis. So, the location of the GPS antenna can be expressed by Equation (Equation 31). In addition, the θ of the measurement vector is an angle based on the slope of the lane, which can be obtained from lane detection. Because this angle is from the slope of the lane concerning the vehicle center axis, it can be obtained as the difference between the current vehicle angle and the current lane angle of the HD map, as shown in Figure 12b. In the ODD environment of this paper, because it is a two-lane movement in each direction, the calculated lane angle from the slope is based on the center line of the HD map. Therefore, the measurement vector $z_{k}$ can be expressed by Equation (Equation 31). Similar to $A_{k}$, the measurement transition matrix $H_{k}$ of the EKF is determined by expressing the Jacobian matrix of ${\hat{x}}_{k}^{\;\bar{}}$ for the function *h*, as shown in Equation (Equation 32).

As mentioned earlier, the information obtained from the RTK module and lane recognition results are sometimes unreliable. Regardless of the RTK module’s accuracy, the position may shift depending on the sensor status or GPS signal reception rate. Lane recognition also differs in recognition rate depending on the surrounding environment, and the lane is only sometimes recognized in all frames. In addition, even if the lane is recognized, the recognition itself could be wrong. Therefore, during autonomous driving, we compared the value estimated by the VPE with the value obtained through the RTK module and lane recognition. If the difference *e* is more significant than a certain threshold, the state of the RTK module or lane recognition is considered abnormal, and the vehicle is immediately stopped as a fallback.

### 5.4. Global Path Planning

For the vehicle to travel to its destination, it is essential to plan the entire path from the start to the end. In general, we need a map of the environment to plan the entire path from the current position to another position. Through this map, we determine where we are presently on the map and plan a path to reach our destination. In this paper, we already have a map as an HD map, and through this, we can plan the entire path. In general, a map is generated using the sensor data, and the vehicle’s current position is identified through this generated map. However, in this paper, because we already have an HD map, we do not need to create a map through separate sensors. In addition, because it uses the RTK module, the current position can be determined directly without any SLAM method.

When the destination is set through the UI, we plan the entire path using the HD map. Because the HD map has the node and link information, we used these to plan the entire path. The nodes refer to points in each section of roads, and each link refers to a set of points between nodes. In our HD map, every node and link has its ID. Each node has its ID and link IDs connected to this node, and each link has its ID and node IDs of where the link starts from and ends. Here, we plan the entire path using the information of the links associated with the node and the information of the nodes from where the link is from and where it ends.

Node $n_{i}\in N$ has an ID $n_{i}^{\mathrm{ID}}$, link IDs $l_{n_{i}}=\left\{l_{n_{i,1}},l_{n_{i,2}},\ldots ,l_{n_{i,m}}\right\}$, and a node point $n_{i}^{P}$. The link $l_{j}\in L$ has its own ID $l_{j}^{\mathrm{ID}}$, from node ID $l_{j}^{FN}$ to node ID $l_{j}^{TN}$, and link points $l_{j}^{P}=\left\{l_{j,1}^{P},l_{j,2}^{P},\ldots ,l_{j,k}^{P}\right\}$. As mentioned above, we can obtain the goal position $p_{g}=\left(x_{g},y_{g}\right)$ through the UI. We assume the current position is $p_{c}=\left(x_{c},y_{c}\right)$. In this case, we first need to determine the link in which we are located. To determine this link, we compared the links in the HD map with the current position $p_{c}$.
(33)$$l_{c}=\underset{l}{\arg\min}\|l^{P}-p_{c}{\|}^{2}$$

Similarly, we can obtain a link $l_{g}$, where the goal position is located.
(34)$$l_{g}=\underset{l}{\arg\min}\|l^{P}-p_{g}{\|}^{2}$$

As mentioned above, we already know the connections between links and nodes. From these connections, we can create paths to reach the target link. Here, we make all paths to reach the target link $l_{g}$ from the current link $l_{c}$, and the shortest path is selected from them.

As shown in Algorithm 1, we plan our global path from $p_{c}$ to $p_{g}$. Firstly, we start from link $l_{c}$, where the vehicle is located. From $l_{c}$, we can obtain the next node ID $l_{j}^{TN}$. We can obtain the links connecting this node from $l_{j}^{TN}$. If we have a way to reach this goal, we can use this process repeatedly to reach the goal link $L_{G}$. In this paper, we adopted a recursive function to repeat this process. We can obtain multiple paths from this function rather than only one path. If we take more than two paths, we choose only one path with the shortest length. Note that we are not concerned about the lane change because our ODD has a two-lane movement in each direction.

After making the shortest path to reach the goal from the above process, we must determine the status of the turn signal light. CAN communication is used to handle the turn signal light of the vehicle; to properly switch on the turn signal light of the vehicle, we need to determine when the turn signal light should be on and the direction of the turn signal light. If the vehicle needs to change in another direction at a crossroad, the vehicle must switch on the turn signal light to let the surrounding vehicles know the direction the vehicle will take.

In the above process, as we have already completed the global path planning of the vehicle, we use this information to determine the state of the vehicle’s turn signal. Because lane change is not allowed in our environment, the turn signal only needs to be turned on in some limited specific situations, such as if a crossroad is present. Crossroads can be easily distinguished by the number of links associated with nodes. If a node has more than one link, we determine that the node is at a crossroads. Although we found crossroads, we may not necessarily have to turn on the signal light at this crossroad. If it is straight from the crossroad, it does not need to turn on the direction indicator. Therefore, we must decide whether the signal light must be turned on or off at the crossroad.

Based on the crossroad node $n_{k}=n_{p_{r,k}^{L}}$, we consider two links to determine whether the turn signal light has to be switched on or off. One is the link $l_{k}=l_{p_{r,k}^{L}}$ before the crossroad node $n_{k}$, and the other is the link $l_{k+1}=l_{p_{r,k+1}^{L}}$ after the crossroad node $n_{k}$. As mentioned above, links have their own points $l_{k}^{P}=\left\{l_{k,1}^{P},l_{k,2}^{P},\ldots ,l_{k,m}^{P}\right\}$ and $l_{k+1}^{P}=\left\{l_{k+1,1}^{P},l_{k+1,2}^{P},\ldots ,l_{k+1,m}^{P}\right\}$. Based on $n_{k}$, we create a straight line based on points of size *b* from the end of $l_{k}$ and take points of size *f* from the beginning of $l_{k+1}$ to create another straight line, as shown in Figure 13. The state of the turn signal light is determined according to an angle θ calculated by these two straight lines. If the magnitude of this angle θ is greater than a certain threshold *h*, the signal light must be turned on, and the sign of θ determines the direction of the turn signal light. In addition, the turn signal light is not only switched on while turning to the curve but must be turned on to indicate other vehicles in the direction that we want to change even before turning the curve. If it is determined that the signal light should be turned on at the next link $l_{k+1}$, it is also turned on for a certain distance *s* from the end of link $l_{k}$ to inform the surrounding vehicles in advance.
**Algorithm 1** Global Path Planning Process**Initialize:** The $l_{c}$, $l_{g}$ from Equations (Equation 33) and (Equation 34) with the $p_{c}$, $p_{g}$. The total path list $p_{k}\in P$ include a link ID $p_{k}^{L}$, a length $p_{k}^{D}$, a success flag $p_{k}^{S}$, a in-depth $p_{k}^{I}$, next links $p_{k}^{NL}$ and their lengths $p_{k}^{ND}$ from the end of the path $p_{k}$. The max depth variable $d_{m}$. The final result path $p_{r}$ with maximum length.**function** Recursive ($p_{k}$)1:**for**$p_{k}^{NL}$**do**2:    **if**
$p_{k,i}^{I}>d_{m}$
**or**
$p_{k,i}^{NL}==l_{g}$
**or**
$p_{k,i}^{NL}$ in $p_{k}^{L}$
**then**3:        **if** $p_{k,i}^{NL}==l_{g}$ **then**4:           $p_{k}^{S}$ = True5:        **else**6:           $p_{k}^{S}$ = False7:        **end if**8:    **end if**9:    $p_{k^{\prime }}=p_{k}$10:    $p_{k^{\prime }}^{L}\overset{\mathrm{add}}{\leftarrow }p_{k,i}^{NL}$11:    $p_{k^{\prime }}^{D}+=p_{k,i}^{ND}$12:    $p_{k^{\prime }}^{I}+=1$13:    Take $p_{k^{\prime }}^{NL}$ and $p_{k^{\prime }}^{ND}$ from the end of the path $p_{k^{\prime }}$14:    $P\overset{\mathrm{add}}{\leftarrow }p_{k^{\prime }}$15:    Recursive($p_{k^{\prime }}$)16:**end for****function** Main ($p_{c}$, $l_{c}$, $p_{g}$, $l_{g}$)1:Take $p_{k,1}$ from the $p_{c}$ and $l_{c}$2:$P\overset{\mathrm{add}}{\leftarrow }p_{k,1}$3:Recursive ($p_{k,1}$)4:**for***P***do**5:    Get total length $t_{l}$ of $p_{k,i}$6:    **if** $p_{k,i}^{S}$ is True **and**
$t_{l}<p_{r}^{D}$ **then**7:        $p_{r}=p_{k,i}$8:    **end if**9:**end for**10:Add the goal information to $p_{r}^{L}$ and $p_{r}^{D}$ from the $p_{g}$ and $l_{g}$

### 5.5. Vehicle Navigation

#### 5.5.1. Planner

We constructed an autonomous driving state machine that defines the vehicle control state, as shown in Figure 14. This state machine represents the control-state flow of a human-intervened or autonomous driving system. Through this state machine, we can efficiently manage the state of the autonomous vehicle and prevent exceptions during autonomous driving. In this paper, the vehicle is divided into three states: a manual state operated by a human, an auto state in which the autonomous driving system operates, and an emergency state in which the vehicle is stopped immediately.

The manual state and the auto state are altered by the human or by applying the run/stop button inside the vehicle, the remote control, or the UI. Additionally, in the auto state, the vehicle state is converted to a manual state when the state machine assumes that the person acquires control of the vehicle, such as a handle, accelerator pedal, or brake pedal is applied (take over). Moreover, the emergency state is switched when applying the emergency button, when an object within the emergency detection range is detected in the autonomous driving system, or when it is determined that an abnormality (fallback) has occurred in the autonomous driving system during the auto state. At this moment, the vehicle’s state is immediately changed from auto to emergency. Then, the system operates an emergency stop by giving the maximum deceleration value until the emergency state is released. To release this state, a person must directly release the emergency state by physically manipulating the UI. The state machine is designed such that a person directly determines the problem of an emergency. After confirming that the situation is controlled, the person releases the emergency state directly.

Inside the box of the auto state in Figure 14, the autonomous driving motion represents the vehicle’s operation during autonomous driving. The auto state of the vehicle is divided into a straight, left turn, right turn, and wait. For the vehicle to be in motion, specific actions are needed: drive, low drive, keep distance, acceleration, deceleration, and stop. Furthermore, driving behaviors are determined according to the above motions. These motions are changed according to the planned global path of the vehicle and the recognized objects. In particular, the wait motion makes the vehicle stop and wait in a traffic situation or maintains the vehicle’s speed of 0 until the stop action is released when the vehicle arrives at the crosswalk. Although there is a crosswalk without any traffic signal, we configured the vehicle to stop at the stop line of the crosswalk. After waiting enough time, the vehicle releases the waiting motion. This waiting time is reinitialized according to the object’s presence near the crosswalk area. Algorithm 2 expresses the process of determining whether an object exists inside or outside the crosswalk area by the crosswalk area of the HD map and the recognized object information. In this algorithm, the wait motion’s time is reinitialized according to the number of intersection points between the straight line drawn along the parallel axis concerning the object and the crosswalk area if an object is recognized.
**Algorithm 2** Crossroad object detection**Input: object pose ($P_{obj}$), each crosswalk polygon point: C[n]****Output: isInside [True/False]**1:Set cross number counter: $c_{cnt}=0$                                                            *Check number of intersections of horizontal line and polygons*2:**for**i=0,1,...,n**do**3:    Set j=(i+1)%n4:    **if** $\left(C\left[i\right].y>P_{obj}.y\right)\;!=\left(C\left[j\right].y>P_{obj}.y\right)$
**then**                                                                                                                 *If object pose is inside y-axis*5:        Set $x=\left(C\left[j\right].x-C\left[i\right].x\right)*\left(P_{obj}.y-C\left[i\right].y\right)/\left(C\left[j\right].y-C\left[i\right].y\right)+C\left[i\right].x$                                                                                                     *Calculate x-axis value of cross point*6:        **if** $P_{obj}.x<x$ **then**7:           $c_{cnt}=c_{cnt}+1$8:        **end if**9:    **end if**10:**end for**11:**if**$c_{cnt}\%2==1$**then**                                                                                              *Odd intersections mean object is inside*12:    **return**
True13:**else**14:    **return**
False15:**end if**

As mentioned above, we defined the action of the autonomous vehicle as follows to implement each motion: the actions are classified as standard drive action running at maximum speed, stop action stopping the vehicle, acceleration action to accelerate the vehicle, deceleration action to decelerate the vehicle, keep distance action to maintain a distance between the autonomous vehicle and the front vehicle, and low drive action running at a specific low speed. Each of the above actions is changed according to the vehicle’s path, surrounding object information, and speed limit information of the road during the autonomous driving motion. The vehicle’s control unit determines the target speed based on the above information.

The vehicle generally travels on the road at a maximum driving speed of 5 km/h. When a specific obstacle or vehicle is present on the driving path, the autonomous vehicle performs deceleration or stops the action to avoid collision with the obstacle or vehicle. The keep distance action maintains a certain distance or, if necessary, stops if any vehicle ahead of us is moving at speed equal to or less than ours. The low drive action is changed when safe driving is required, such as where a section of garbage collection or intersections where the vehicle’s speed is lowered.

Figure 15 shows the actions of the autonomous driving system classified according to the current location and speed of a vehicle or any object recognized by LiDAR and cameras. The green vehicle is equipped with the autonomous driving system shown in this paper, and the surrounding red vehicles are the vehicles recognized by our autonomous vehicle. Suppose a part of the red vehicle’s body enters the lane where the autonomous vehicle is traveling. In that case, this vehicle is specified as a corresponding vehicle. The autonomous vehicle changes the action according to the distance from the corresponding vehicle existing on the current lane and does not consider other recognized vehicles. If the corresponding vehicle comes within 3 m, it is changed to ① Emergency Stop; if it enters within 6 m, ② Stop, and if it enters within 10 m, it is changed to ③ Keep Distance action. We use the object-oriented bounding box (OBB) [47] method to determine whether a recognized vehicle and an autonomous vehicle are in the same lane. This method determines whether the currently recognized vehicle exists in the lane by comparing the area considering the width of the lane in which the autonomous driving vehicle currently drives and the bounding box area of the recognized vehicle.

Figure 16 presents an example of how to calculate the distance between vehicles in Figure 15, Algorithm 3 shows how to find the link where the autonomous vehicle is located. Algorithm 4 expresses how to determine whether the recognized vehicle is in the same lane based on the Algorithm 3. The values used in the equation are the global pose $P_{bumper}$ of the front bumper of the autonomous vehicle, HD map link information $L_{i}$ of the lane where the autonomous vehicle is currently driving, and a pose of the recognized vehicle $p_{recog}$. When calculating the distance between the recognized vehicle and the autonomous vehicle, the distance from the front bumper of the autonomous vehicle to the rear bumper of the recognized vehicle is calculated using bounding box information, and the action of the autonomous vehicle is changed accordingly.
**Algorithm 3** Find driving link index**Input: vehicle pose ($P_{v}$), each link point: L[n]****Output: Current HD map link index (Id)**1:**for**i=0,1,…,n**do**2:    **if** $∥\vec{L\left[i\right]P_{v}}∥<∥\vec{L[i]L[i+1]}∥$ **then**                                                                                                          *If vehicle inside link boundary*3:        **if** $\vec{L\left[i\right]P_{v}}\cdot \vec{L[i]L\left[i+1\right]}>0$ **then**                                                                                *If the vehicle’s position is in the direction of the link*4:           Id=i5:           **return** Id6:        **end if**7:    **end if**8:**end for**

**Algorithm 4** Calculate distance from recognized object
**Input: vehicle bumper pose ($P_{bumper}$), recognized object pose ($P_{recog}$), each link point: L[n]**

**Output: distance (*d*)**

1:Set each link point (x,y): $L_{k}$2:Get vehicle link vector: $l_{i}=\vec{L[i]L[i+1]}\;\left(i\;from\;Algorithm\;1\right)$                                                                                                 *Calculate distance vehicle to next link*3:

$d_{1}=∥proj_{l_{i}}\vec{P_{bumper}L\left[i+1\right]}∥$

4:Get object link vector: $l_{j}=\vec{L[j]L[j+1]}\;\left(j\;from\;Algorithm\;1\right)$                                                                          *Calculate distance object backside position from link*5:

$d_{2}≃∥proj_{l_{j}}\vec{L\left[j\right]P_{recog}}∥-d_{recog}/2$

6:

$d_{3}=\sum_{k=i}^{j-1}∥\vec{L[k]L[k+1]}∥$

7:Distance vehicle bumper to recognized object: $d=d_{1}+d_{2}+d_{3}$                                                                                                                      *Calculate links distance*8:
**return**
*d*



#### 5.5.2. Coordinator

According to the vehicle state determined by the planner through the recognized objects in real-time, the autonomous vehicle drives by following the global path. To follow the planned global path, we aimed to maintain a proper lane based on the current pose of the vehicle, using the recognized lane information and the HD map links information. Suppose the vehicle poses information obtained from the VPE and the HD map information match. In that case, we convert the linking points of the HD map into the vehicle’s local coordinate system using the vehicle pose information and obtain the exact path of the vehicle. However, as mentioned earlier, the VPE becomes unstable depending on the GPS, odometer, and lane recognition errors. Therefore, in this paper, the vehicle is driven only when the vehicle pose information obtained from the VPE is valid.

If the vehicle poses information obtained from the VPE is valid, we created local linking points from the global linking points in the HD map using the transition matrix obtained from the vehicle pose to match the vehicle coordinate system. Because the linking points of the HD map are discrete points with a minimum interval of 3 m, we express these points as a continuous path through interpolation between the linked points to drive the vehicle smoothly. In this paper, we generated accurate and smooth continuous vehicle paths through the monotone cubic Hermite interpolation [48] method. The linear interpolation method is improper to drive because it creates a step-shaped path with a sharp angle at each corner. The cubic spline interpolation [49] method creates a much more natural path than the linear interpolation method. However, as shown in Figure 17a, it is not appropriate for this paper because the interpolation between specific links may fluctuate predominantly. Therefore, we adapt the monotone cubic Hermite interpolation method to create a smooth and fluent driving path that maintains the monotony of the link, as shown in Figure 17b.

#### 5.5.3. Executor

An executor controls the vehicle and drives it according to the planned path calculated by the coordinator. The vehicle is controlled with the steering value and acceleration value transferred via the CAN communication after calculating the determined action, local path, vehicle poses information, and vehicle encoder information.
(35)$$\delta \left(t\right)=\theta \left(t\right)-\theta_{p}\left(t\right)+\tan^{-1}\left(\frac{k_{p}e_{f_{a}}\left(t\right)+\int k_{i}e_{f_{a}}\left(t\right)dt}{v(t)}\right)$$

In this paper, we use the Stanley model [50] to determine the direction of the vehicle by controlling the steering of the vehicle. Equation (Equation 35) shows the calculation of the current steering angle δ. The vehicle state δ is generated by the current speed *v* from the vehicle’s encoder and heading θ estimated from the VPE. In the Stanley method, $e_{f_{a}}$ is the distance from the front wheel of the vehicle to the nearest path point $\left(c_{x},c_{y}\right)$, θ is the heading angle of the vehicle, and $\theta_{p}$ is the heading angle at the nearest path point $\left(c_{x},c_{y}\right)$. Unlike the original Stanley model, in this paper, we add an integral term $\int k_{i}e_{f_{a}}\left(t\right)dt$ to compensate for the steering δ of the vehicle at the steep corner path or the continuous corner path. The integral term accumulates $e_{f_{a}}$ during a specific time *T*. The larger this integral term, the greater the δ, which causes more enormous steering changes. However, because the continuous accumulation of the integral term causes significant changes even when not required, we use a clamped anti-windup technique to prevent saturation of the integral term. Finally, $k_{p}$ and $k_{i}$ are hyperparameters predefined by the user. The values of these hyperparameters are determined to hold the collected garbage stably. These values differ greatly depending on the vehicle and the vehicle’s condition. Therefore, these values are determined heuristically by driving each section of the experimental environment.

We designed the PID controller, as shown in Figure 18, for the longitudinal control of the vehicle and controlled the target speed to execute the action. Then, we must know the relationship between the control value and acceleration/deceleration for the execution. We performed the acceleration/deceleration test based on feeding static control values to get the relationship. Here, we limited the control value based on the maximum and minimum values from the test within the system. Furthermore, we add the anti-windup technique to prevent saturation and initialize the integral term in the PID controller.

In autonomous driving, the vehicle’s maximum speed is determined by the maximum speed set in the HD map for the current driving route. However, the planner’s state changes according to the surrounding environment, and the desired speed of the vehicle changes accordingly. Accordingly, vehicle accelerator and brake control are performed through PID control based on a difference between the target speed that the desired speed that is limited to the maximum speed, and the current speed.

### 5.6. Fallback

In the case of an ordinary vehicle, the vehicle’s dashboard displays information about the state of the vehicle. Similarly, the autonomous driving system in this paper also displays the status of the autonomous driving system on the UI so that the user can be informed about the autonomous driving status. In the case of autonomous vehicles, various types of sensors are used to recognize the surrounding environment. We used radar, LiDAR, cameras, and ultrasonic sensors for the vehicle. If any of these sensors fail, safety problems may occur while the vehicle is being driven. Therefore, we added a fallback function that detects the current state of these sensors in an autonomous driving system and copes with it. We consider situations in which sensors cannot acquire sensor data due to sensor connection, sensor failure, poor communication, and data acquisition failure due to sensor occlusion. When the above problem occurs, even with only one sensor, we programmed it to notify the user visually and audibly through the UI display to stop the vehicle immediately. However, the failure to acquire sensor data can often occur due to connection failures. In this case, it was not related to sensor malfunction. Moreover, if the connection is cut off for a short time and expected data are acquired again, it is difficult to determine that there is a problem with the sensor. This case is considered only when the data acquisition from the sensor fails for a certain period; it is recognized as a sensor problem, and fallback is performed.

Apart from the state of the sensor for recognizing the environment, it is also essential to check the connection state with sub-controllers connected to the vehicle. Because the sub-controllers move the vehicle or operate the buttons, the autonomous driving system can operate normally when the sub-controllers are in a normal state. These sub-controllers transmit their operating states through CAN communication. The values retrieved from the CAN communication can be used to check whether the sub-controller state is currently operating in a normal state, the state of the steering, brake motors, and the emergency state from the UI. Through this information, based on the status of the sub-controllers, an emergency stop operation of the vehicle is performed if it is determined that autonomous driving of the vehicle is not possible.

## 6. Experiments

To experiment with the autonomous driving system proposed in this paper, we chose a prior experimental area, as shown in the left column of Figure 19, similar to the ODD environment we already defined. According to our ODD, the residential area contains both people and vehicles. Therefore, it is dangerous to experiment in the target area before verifying the actual performance of the autonomous driving system. Therefore, we conducted our experiments in a controlled environment that replicates real-world scenarios. This experimental area contains broad roads, such as expressways marked with a red box in the left column of Figure 19. The remaining section contains a road resembling a city or a semi-urban area consisting of two lanes moving in each direction, similar to the ODD environment.

After experimentation, we extended our developed vehicle to the residential area we targeted in Section 3. As shown in the right column of Figure 19, we composed an HD map based on the target ODD environment. However, since our autonomous vehicle is currently in the development stage, there is a possibility that the vehicle can act unexpectedly, which may lead to unintended dangerous consequences. Therefore, we established an additional safety plan for operating our autonomous vehicle in an actual residential area. As shown in Figure 20, we placed additional vehicles and safety guards while operating the autonomous vehicle. A leading vehicle in front of the autonomous vehicle, which precedes the route in advance, notifies the autonomous vehicle in operation and checks for significant risks on the route simultaneously, while a control vehicle oversees the status and surrounding conditions of the autonomous vehicle and sends an emergency signal remotely if needed. There will always be two security guards next to the autonomous vehicle, imitating the garbage collection task and pressing the emergency stop button on the side of the vehicle as shown in Figure 2 when any unprecedented situation occurs. Moreover, additional safety guards are also deployed at certain sections and intersections. These security guards will inform the surrounding vehicles and people that the autonomous vehicle is in operation and will play a role in controlling the surrounding vehicles and people in case of an emergency.

The vehicle’s core software system was configured based on ROS. All hyperparameters required for recognition, steering, and speed control were first set as initial values based on the simulation environment and then empirically adjusted while driving in various conditions such as road obstruction, emergency stop, and uphill/downhill in the target environment. Although the processing time for the modules of the system is different for each module, the communication cycle of the entire system is set to 100 ms, accordingly.

In this paper, we tested a total of six items. In addition to position, steering, and speed, which are essential functions for vehicles to drive autonomously along a route, experiments were also conducted to recognize people and control speed. In addition, an experiment was conducted to see if the emergency stop buttons additionally installed for safety worked as expected. Finally, an experiment was conducted to present whether the developed vehicle could support garbage collection tasks.

### 6.1. Position Result

Knowing the position of a vehicle is one of the most critical factors in the autonomous driving of a vehicle. In this paper, we adjusted VPE to estimate the vehicle’s position. Figure 21 shows the planned path based on the HD map in the section with the greatest curve degree among the entire route in the experimental area, the position acquired from the RTK module, and the position estimated by the VPE. As seen from the above results, the planned path (blue) is not a continuous one because the paths in the HD map are discrete. To follow this planned path, the VPE estimated the vehicle’s pose continuously. Furthermore, as shown in Figure 21, the VPE (yellow) shows less fluctuated location results than the RTK module (red) even though the RTK module generated precise locations. Based on the planned path, the variance with the RTK module was 0.0881, and the variance with the VPE estimate was 0.0835.

### 6.2. Steering Result

In this experiment, we set $k_{p}$ to 0.4 and $k_{i}$ to 0.2 to calculate the steering angle. In addition, as mentioned earlier, we applied the anti-windup technique to an integral part of Equation (Equation 35). The saturation limit in this equation was set to [0.05,12.0].

Figure 22a shows the graph of the desired steering angle and the current steering angle of the front wheel. Figure 22b shows the graph of the desired handle angle and the current handle angle of the steering wheel. The x-axis of these graphs represents time in units of 100 ms, the y-axis in Figure 22a represents the steering angle of the front wheel, and the y-axis of Figure 22b represents the rotation angle of the handle. As shown in Figure 22a, the steering angle of the current front wheel (red) is almost the same as the desired steering angle of the front wheel (blue) every time. However, the desired steering angle is more varied than the actual one. This variance is because the value calculated every hour in Equation (Equation 35) is calculated independently over time and does not consider the previous steering angle. In addition, the actual steering angle cannot immediately follow the calculated desired steering angle because there is a limit to the speed at which the actual vehicle’s handle turns physically. Thus, for these reasons, we can observe a graph in which the current steering angle has less variance than the calculated steering angle, as shown in Figure 22a. Figure 22b also shows how the current handle angle (red) follows the desired handle angle (blue) with a slight delay, as mentioned above. Although there is a slight delay in the current steering angle according to the control command, we suppose that there are no problems with the steering control because the vehicle drives very slowly at a speed of under 5 km/h, which is enough to ignore this delay in the ODD environment.

### 6.3. Velocity Result

As shown in Figure 18, the vehicle speed is controlled using a PID controller. To calculate the deceleration/acceleration of the vehicle in this experiment, we set the PID controller variables $k_{p}$, $k_{i}$, and $k_{d}$ to 1.5, 0.1, and 0.1, respectively. In addition, the saturation limit value of the PID controller was set to [−4.0,2.0], and the saturation limit value of the integral term was set to [−8.0,8.0].

Figure 23 shows the graph of the acceleration values calculated from the PID controller and the actual vehicle speeds. The x-axis of these graphs represents the time in ms units. The y-axis of the first row and second row in Figure 23b represent the acceleration in m/s^2^ units and the speed in m/s, respectively. In the case of Figure 23a, they are expressed together to see the changes in acceleration and velocity over time, even if the units of the actual y-axis are different. Here, we can see that the vehicle’s speed changes according to the increase or decrease in acceleration.

In the case of ordinary vehicles, they start moving at a slower speed when we set the vehicles into drive mode and release the brake paddle. In our vehicle, however, the vehicle is immobile even after releasing the brake paddle. Therefore, as shown in Figure 23, the acceleration and speed overlapped simultaneously. Because the vehicle was driven less than 5 km/h, the acceleration was controlled with a maximum of 2 m/s^2^ for safety reasons. In addition, to stop the vehicle as quickly as possible in an emergency, we provide the vehicle with a maximum deceleration of −4 m/s^2^ to stop the vehicle, as shown in Figure 23. In an emergency, as shown in the graph above, the vehicle speed immediately converges to 0 m/s to make an emergency stop.

### 6.4. Human Recognition Result

Object recognition is one factor to check when driving an autonomous vehicle. There are several methods to check object recognition performance, but this paper focused on recognizing the object in front of the vehicle. The objects visible in the vehicle’s driving path are the most important to pay attention to when driving autonomously. In particular, large objects such as vehicles have no problem recognizing them, but humans are more challenging to recognize because they are relatively tiny and rushed. We experimented with assuming a person appeared in front of the vehicle during autonomous driving. As shown in Figure 24, for safety, the experiment was initially conducted using a model resembling a human, and later, a natural person intervened in the experiment. Of course, safety personnel were always around the vehicle to press the emergency stop button at any time during experiments. During experiments, we produced various conditions while varying the duration from a minimum of 2 m to a maximum of 20 m. In this experiment, we are eager to check that the autonomous vehicle recognizes an object in front and adjusts its speed to not collide with the object. To confirm this, we calculated the time to collide when an object is recognized with the object’s speed and the vehicle’s speed. In Figure 25, the x-axis represents the number of experiments, and the y-axis represents the time until the object collides. For each experiment, the collision time was recorded from when an object was first recognized until the object left the vehicle’s recognition range, which was expressed as a box plot. As shown in Figure 25, the vehicle adjusts its speed gently if there is ample time for collision and more rapidly if there is not enough time when the vehicle recognizes a person in front while performing autonomous driving. In this experiment, we performed a total of 93 human recognition tests; at this time, the median value for the time to collide was about 18.5 seconds, and we could also confirm that the autonomous vehicle recognized a human without failure during the experiment and stably adjusted the speed at the same time.

### 6.5. Emergency Button Response Result

As mentioned above, there are four emergency stop buttons to apply an emergency stop manually. The emergency stop buttons are between the driver’s seat and the passenger seat, on each side of the cabin at the rear, and the remote controller. Here, we checked the response speed of all the emergency stop buttons. Figure 26 shows the signal when each emergency stop button is pressed and the corresponding emergency status signal. Additionally, Figure 26b shows when each emergency signal is flagged. In these graphs, the x-axis represents time in units of ms, and the y-axis represents a binary signal, with one being high and zero being low. When the emergency state signal is activated, the vehicle enters the emergency state and stops with maximum deceleration simultaneously, as shown in Section 6.3. Therefore, as shown in Figure 26a, it can be confirmed that the emergency status signal is activated immediately when the signal from the emergency stop buttons is triggered. The emergency state persists unless the user releases it; hence, it can be confirmed that the emergency state signal is not deactivated even if the signal from the emergency stop buttons is triggered off. Because the emergency state must be released to continue the experiment, the user must manually manipulate the UI to release the emergency stop state and conduct the experiment.

### 6.6. Work Efficiency Result

Our ultimate goal in developing this vehicle is to reduce the burden on working costs while maximizing work efficiency. As mentioned earlier, garbage collection in residential complexes where we are targeting involves two workers employed together to collect garbage, one in charge of driving and the other in charge of loading garbage into vehicles. However, suppose the garbage collection vehicle is capable of autonomous driving by the worker who loads the garbage. In that case, this vehicle can substitute the other worker in charge of driving, thereby reducing the burden of work costs and opening a broad scope for the application of autonomy in vehicles. Regardless of the application and its benefits, when such an autonomous vehicle is introduced, it becomes necessary to test whether it has proper applicability for one worker and the autonomous vehicle to collect garbage compared to two workers to collect garbage in terms of efficiency. Even though we proposed the autonomous driving vehicle, it cannot adapt this vehicle regardless of cost savings if they fall far behind in terms of efficiency compared with two workers collecting the garbage.

In this paper, we measured the work efficiency through the total work time it takes to collect garbage within the actual residential area, as shown in Figure 27. Hence, the above experiments in the experimental area were also performed naturally. Table 1 shows the total time taken to manually and automatically collect garbage in the ODD area. Here, manual means that two workers typically perform the garbage collection task, and automatic means that one operator performs garbage collection work with the autonomous vehicle. Garbage collection tasks were performed ten times, each manually and automatically. As shown in Table 1, the manual operation took less task time than the automatic operation. This table shows that autonomous driving becomes relatively time-consuming due to various variables, such as the difference between human driving and driving through an autonomous system and vehicle operation time. However, as shown in the table, it can be seen that there is not a significant difference in time between manual and automatic operation. Moreover, in terms of work efficiency, it was shown that it took only about 16% longer than when working manually.

## 7. Conclusions

In this paper, an autonomous vehicle and its autonomous driving system for garbage collection in residential areas were developed. We defined the ODD for autonomous vehicles, selectively configured the entire hardware and software, and conducted experiments to verify its functionality in experimental and actual domains. Experiments were conducted on location estimation, steering control, vehicle speed control, human perception, emergency stop function, and work efficiency to confirm that the developed vehicle can drive in residential areas. Based on these experimental results, this paper demonstrates that the vehicle can smoothly drive autonomously within the ODD environment we set up. In addition, measuring the efficiency of work in an actual residential area showed that performing garbage collection tasks together with autonomous vehicles is sufficiently efficient.

However, the limitations presented in this paper are also clear:In the case of the ODD we designed, it is constrained because it is designed only for the particular purpose of garbage collection based on the residential complex. The vehicle’s speed must be shallow, and the driving range is simple, two lanes round trip. In the case of a general vehicle, applying it to various vehicles is challenging because it does not operate based on such an ODD.The autonomous driving of current vehicles relies only on the HD map information. Of course, the HD map configured based on the target environment is very accurate. However, suppose the vehicle loses its position beyond a certain threshold for any reason in this HD map. In that case, the position cannot be recovered, and the vehicle must make an emergency stop.The scope, level of perception, and judgment of the surrounding environment are low. An autonomous vehicle includes driving along the lane, recognizing the surrounding environment, and making decisions based on this. The system of the currently developed vehicle has a straightforward structure that judges its behavior according to the distance of the recognized object. Higher levels of self-driving vehicles require a high degree of judgment, not only recognizing objects but also recognizing and predicting situations in the surrounding environment to determine the final behavior of the vehicle.

## 8. Future Works

In order to overcome the limitations mentioned above in this paper and develop more advanced autonomous vehicles, it is first necessary to define a broader range of ODD. Since garbage collection is performed not only in residential areas but also in industrial complexes and other areas, there is a need to expand the scope of autonomous driving to other areas by defining the ODD proposed in this paper in a broader range. In addition, even if an HD map is provided, SLAM technology must be applied to configure a map based on sensor information and determine the location through which to perform autonomous driving more accurately. Because environmental conditions are constantly changing without notice, the configured HD map may not match the current environment. In this case, if only the built HD map is used, unexpected results such as incorrect estimation or accidents may be obtained in the worst case. A map creation module based on sensor data supports preventing these accidents and more accurate and safe driving. It is possible to perform more accurate vehicle location estimation by integrating it with the existing HD map. Finally, the awareness and judgment of systems mounted on vehicles must be improved. In addition to simple object recognition, it is necessary to recognize the situation around the current vehicle through scene understanding so that appropriate vehicle behavior can be planned. Since recent deep learning studies rapidly develop these cognitive technologies, it is possible to perform a higher environmental awareness by applying these technologies. However, even if we are aware of the situation, making a judgment based on it is another story. Even in the same situation, the result of judgment varies depending on a person by person. So how to make a judgment is also one of the significant factors. Since the basis for such judgment can be greatly influenced by legal and ethical essentials, it is necessary to design the judgment module with care.

## Figures and Tables

**Figure 1 sensors-22-09094-f001:**
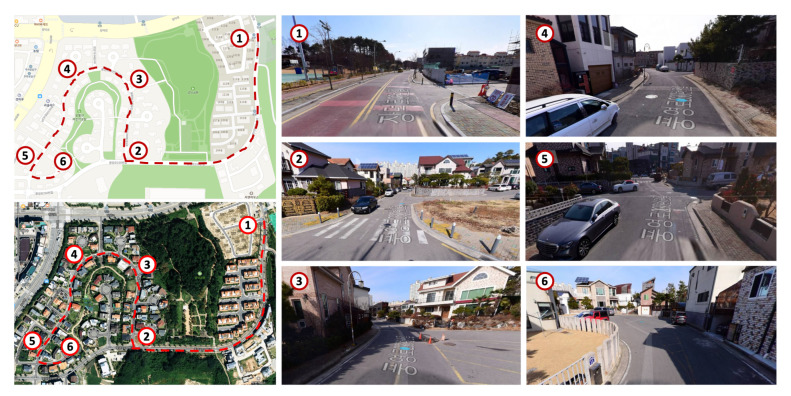
Specific routes in a residential area.

**Figure 2 sensors-22-09094-f002:**
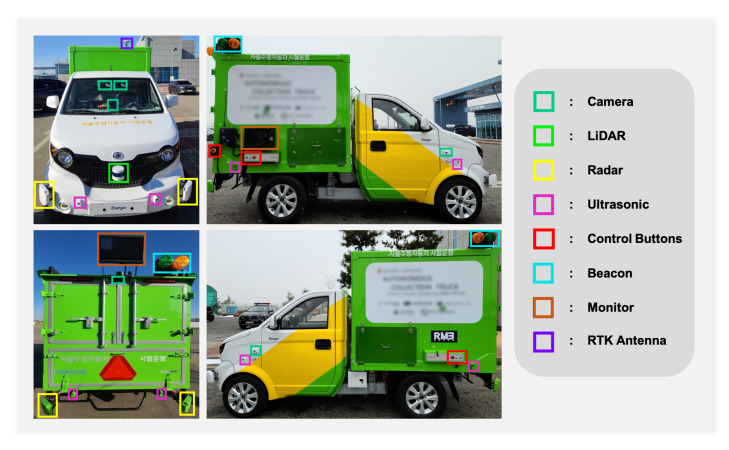
Hardware placement.

**Figure 3 sensors-22-09094-f003:**
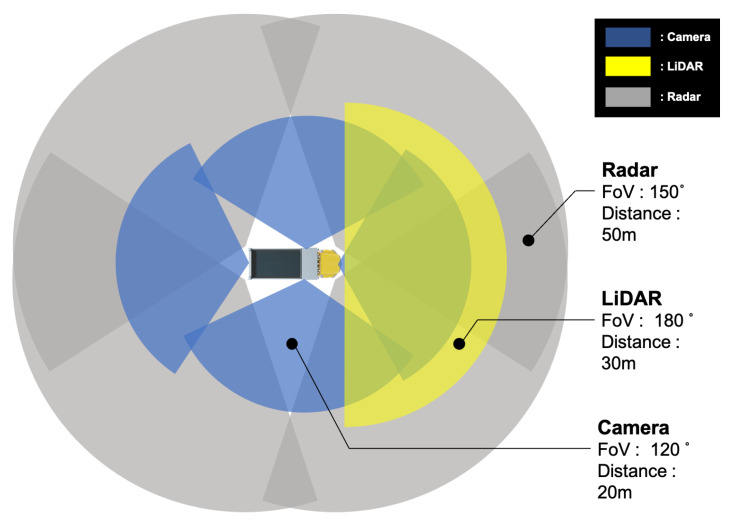
Sensor coverage.

**Figure 4 sensors-22-09094-f004:**
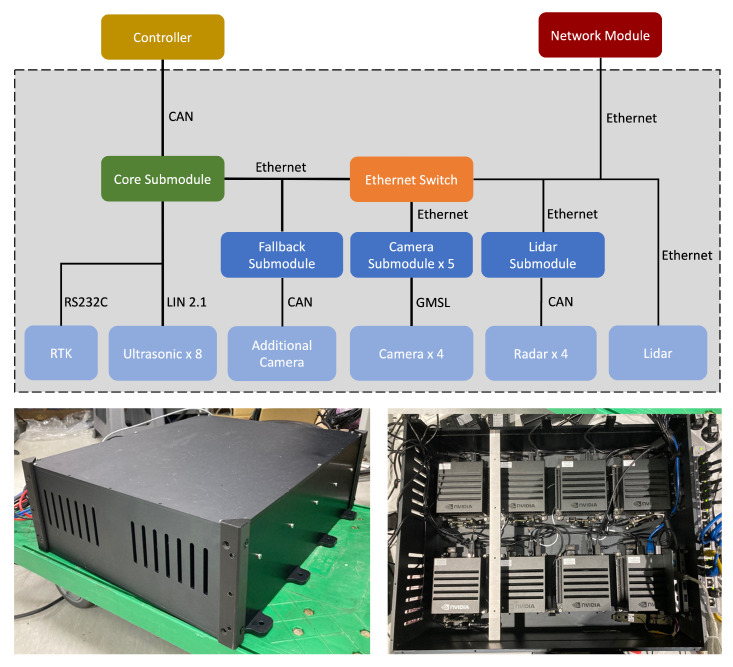
Modulebox structure and actual feature.

**Figure 5 sensors-22-09094-f005:**
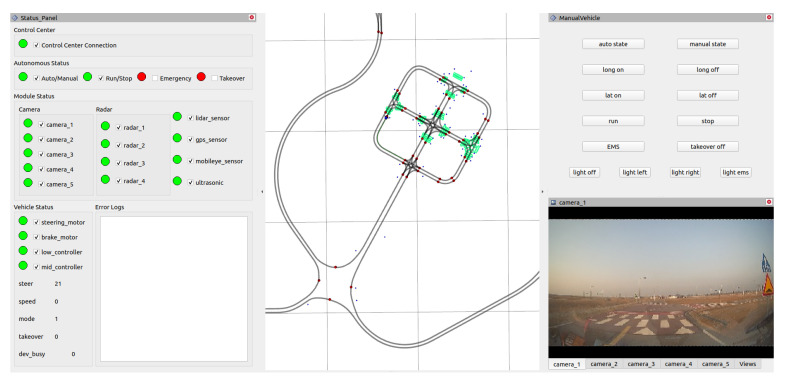
User interface.

**Figure 6 sensors-22-09094-f006:**
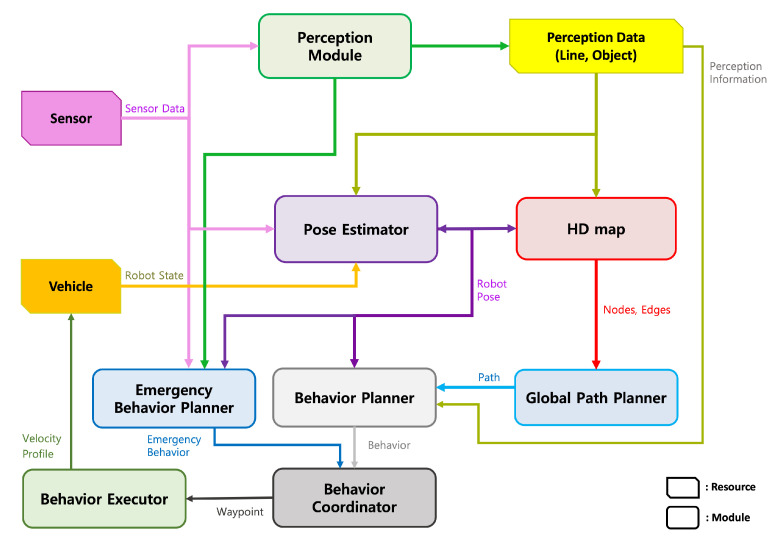
Overall System.

**Figure 7 sensors-22-09094-f007:**
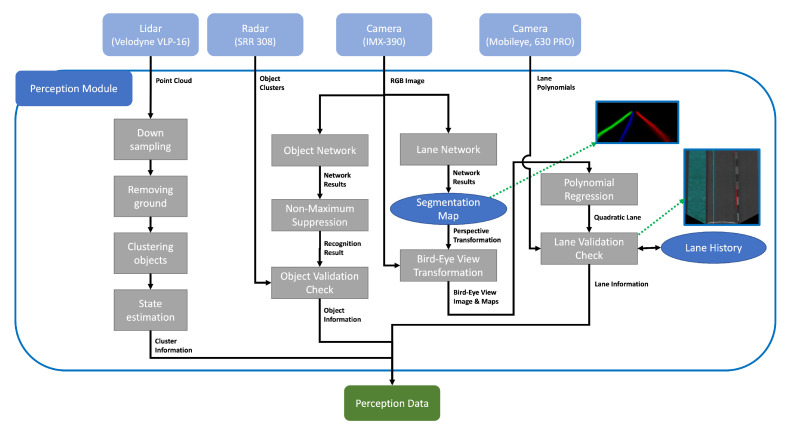
Perception module.

**Figure 8 sensors-22-09094-f008:**
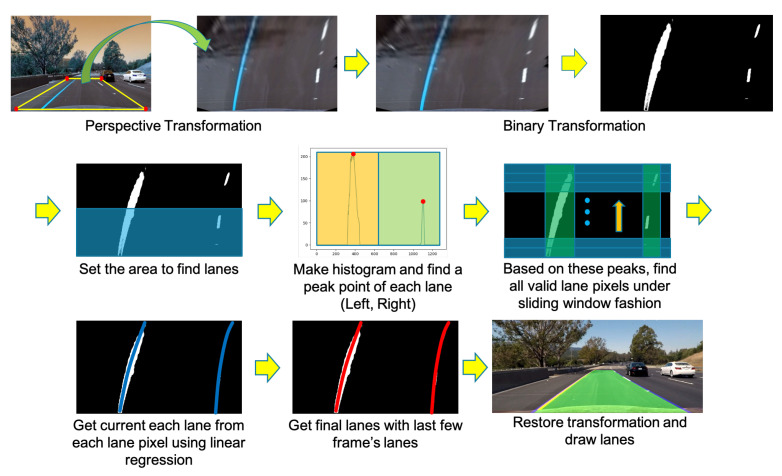
Lane recognition process using binary images.

**Figure 9 sensors-22-09094-f009:**
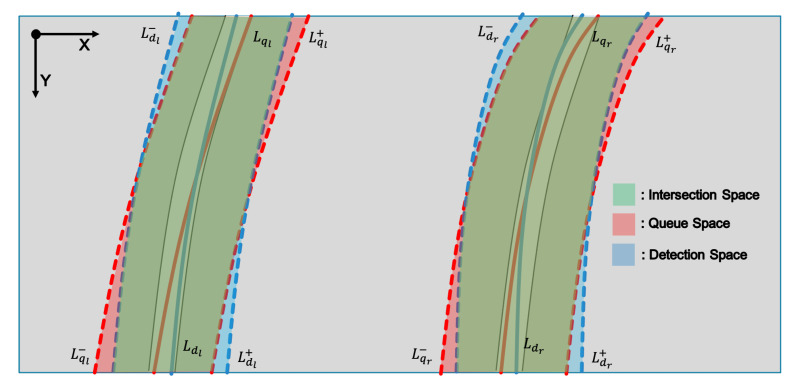
Lane IoU example.

**Figure 10 sensors-22-09094-f010:**
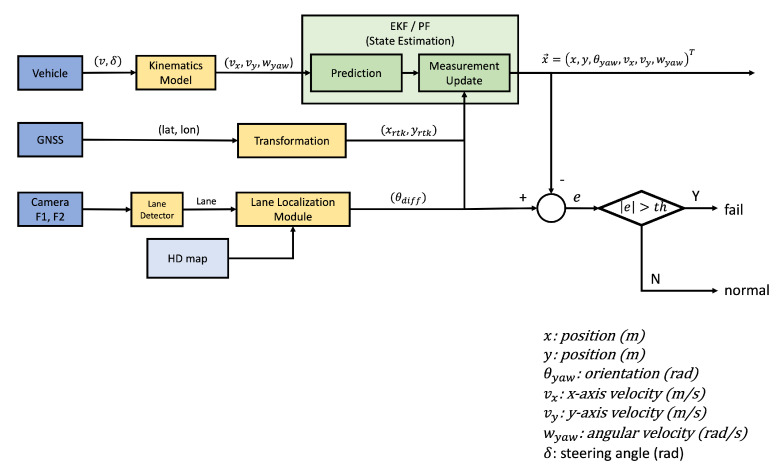
Vehicle pose estimation.

**Figure 11 sensors-22-09094-f011:**
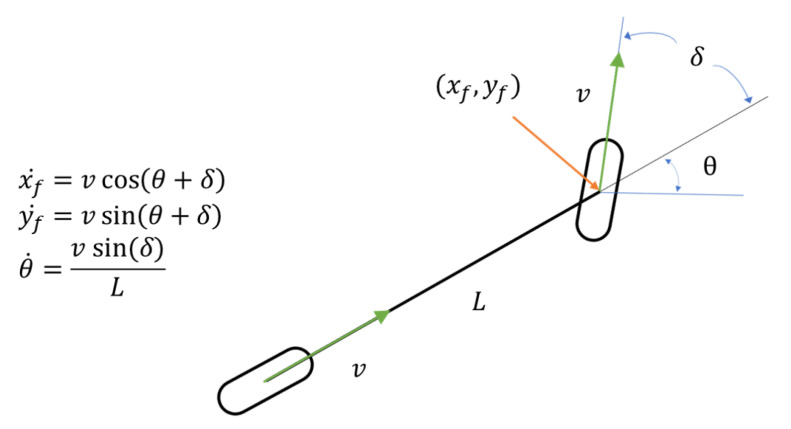
Vehicle kinematics.

**Figure 12 sensors-22-09094-f012:**
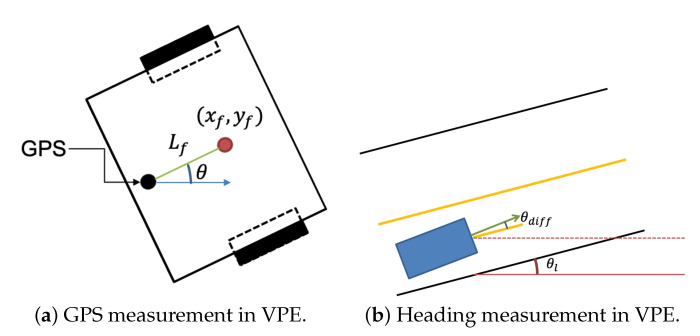
Measurements in VPE.

**Figure 13 sensors-22-09094-f013:**
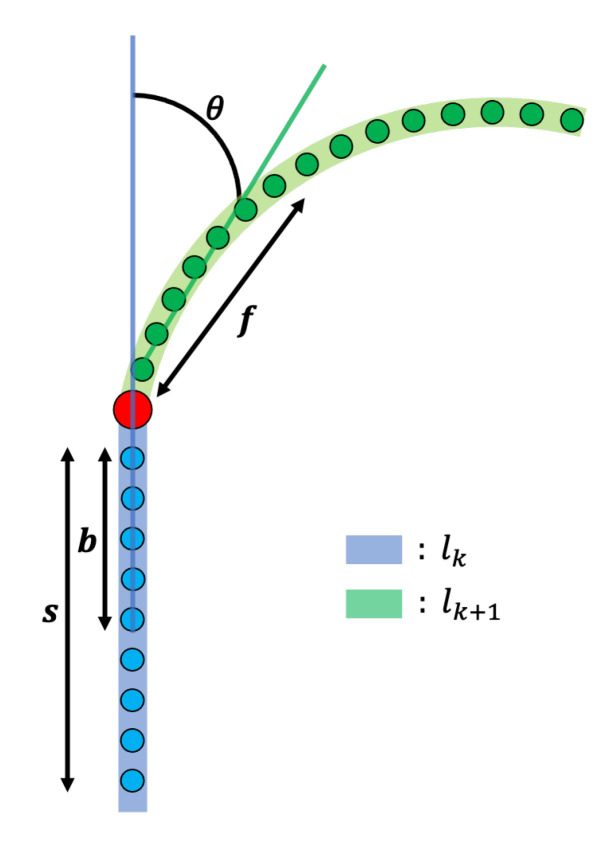
Calculation of θ.

**Figure 14 sensors-22-09094-f014:**
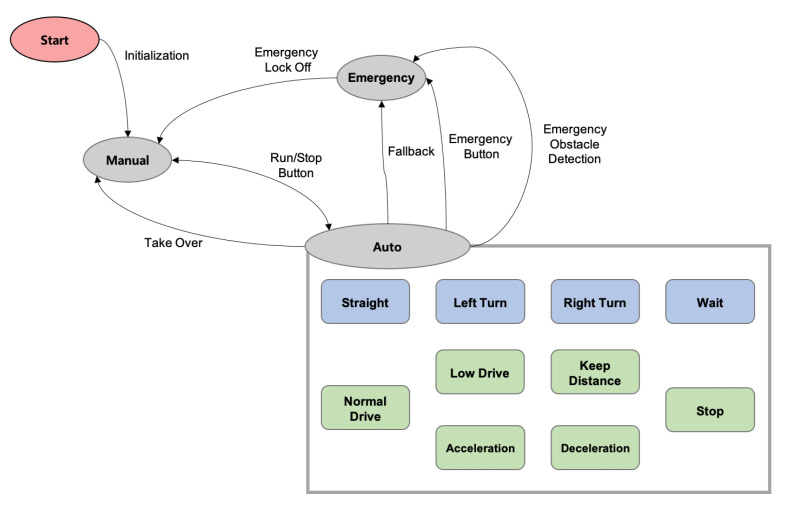
Motion state machine.

**Figure 15 sensors-22-09094-f015:**
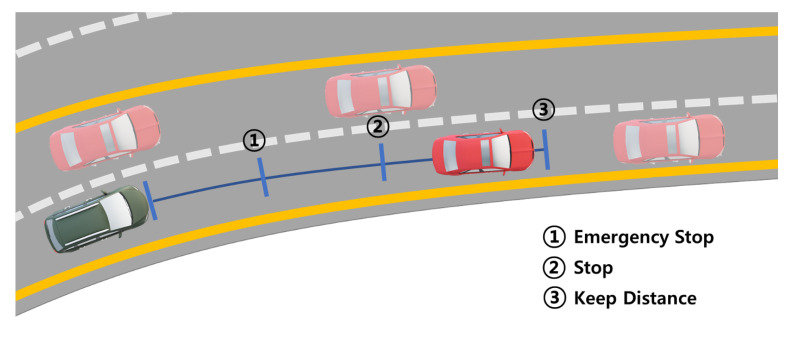
Case of long-control state.

**Figure 16 sensors-22-09094-f016:**
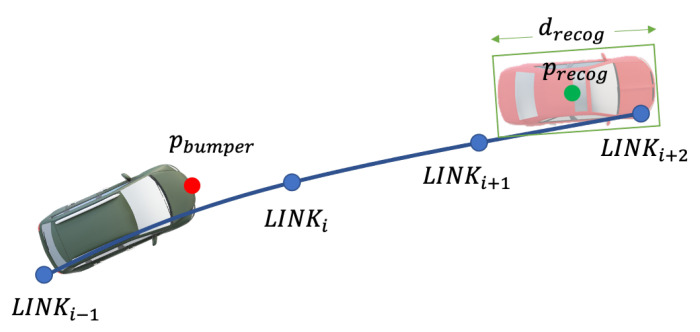
Discrimination-confrontation vehicles.

**Figure 17 sensors-22-09094-f017:**
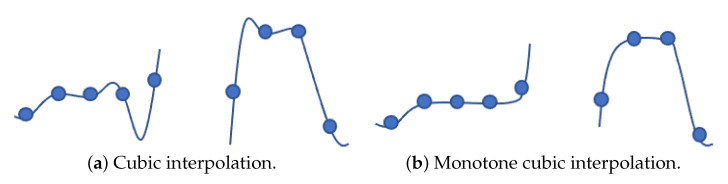
Link points interpolation.

**Figure 18 sensors-22-09094-f018:**
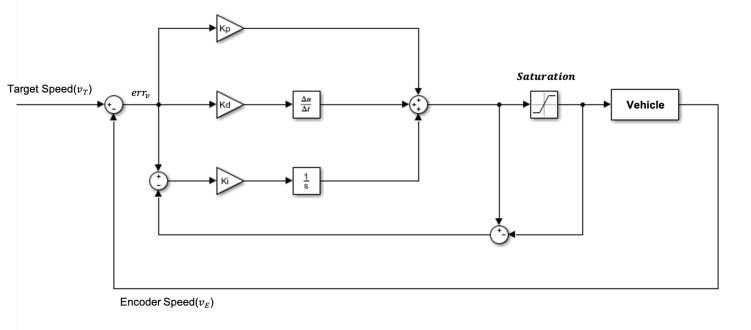
Speed controller block diagram.

**Figure 19 sensors-22-09094-f019:**
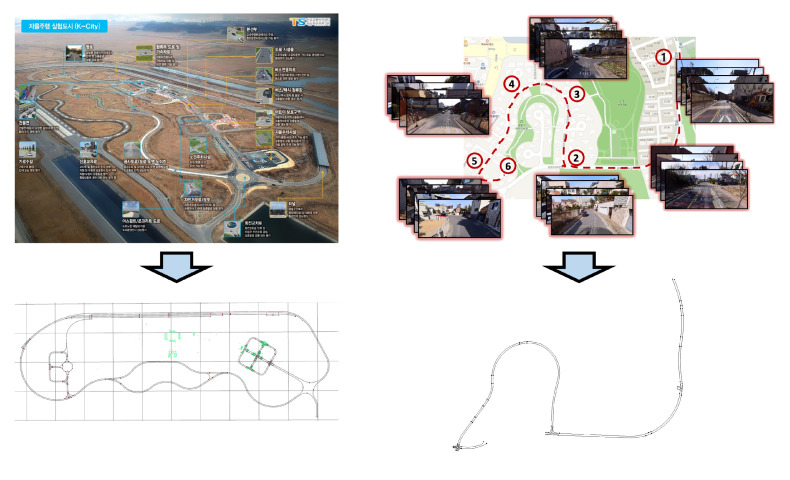
Experimental and actual residential area.

**Figure 20 sensors-22-09094-f020:**
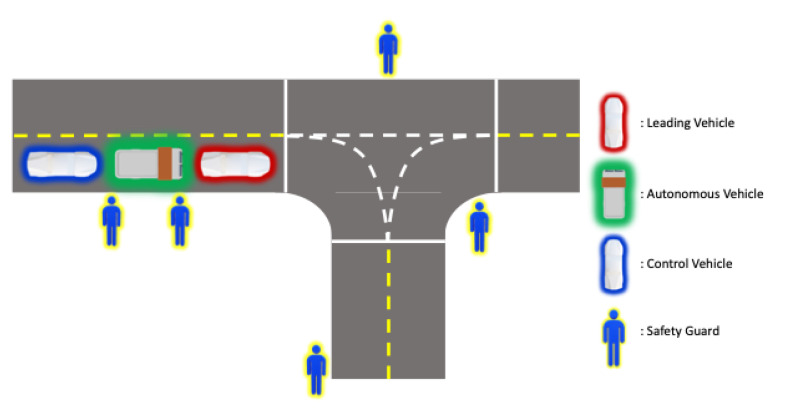
Safety plan.

**Figure 21 sensors-22-09094-f021:**
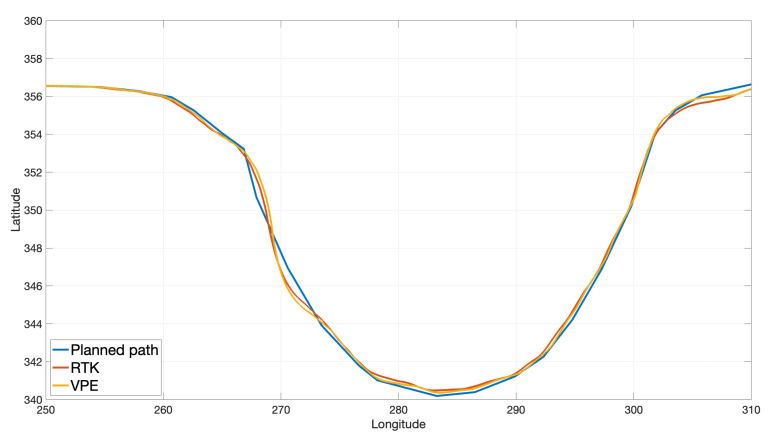
RTK positions and estimated positions.

**Figure 22 sensors-22-09094-f022:**
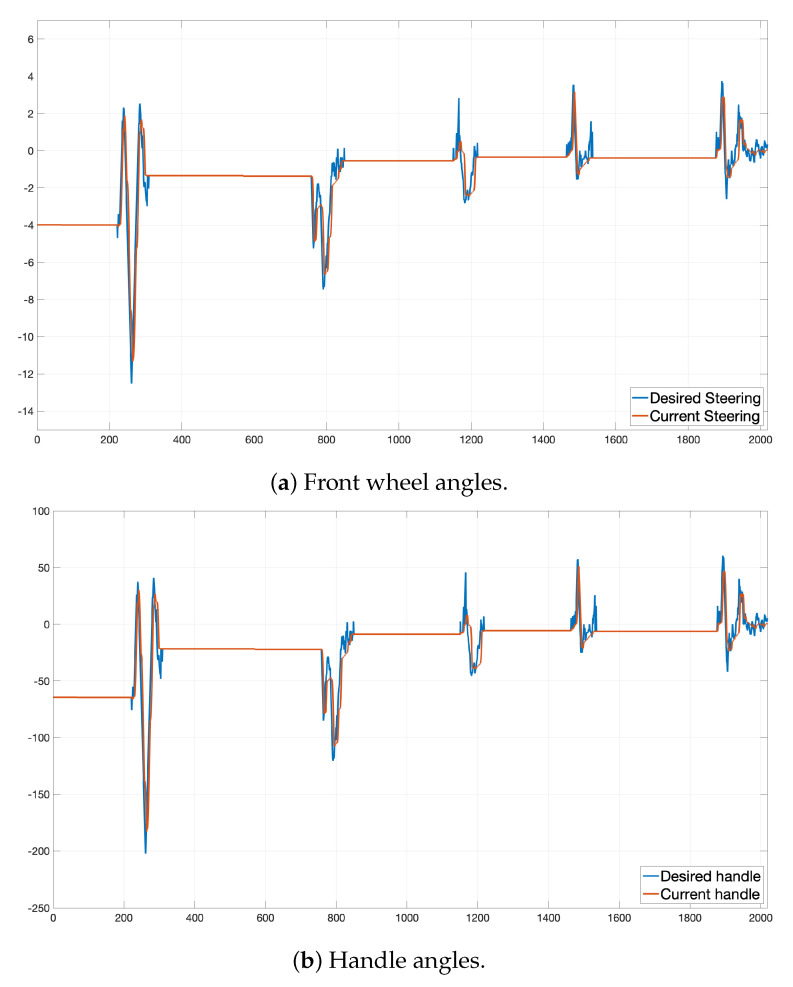
Desired and actual front wheel & handle angles.

**Figure 23 sensors-22-09094-f023:**
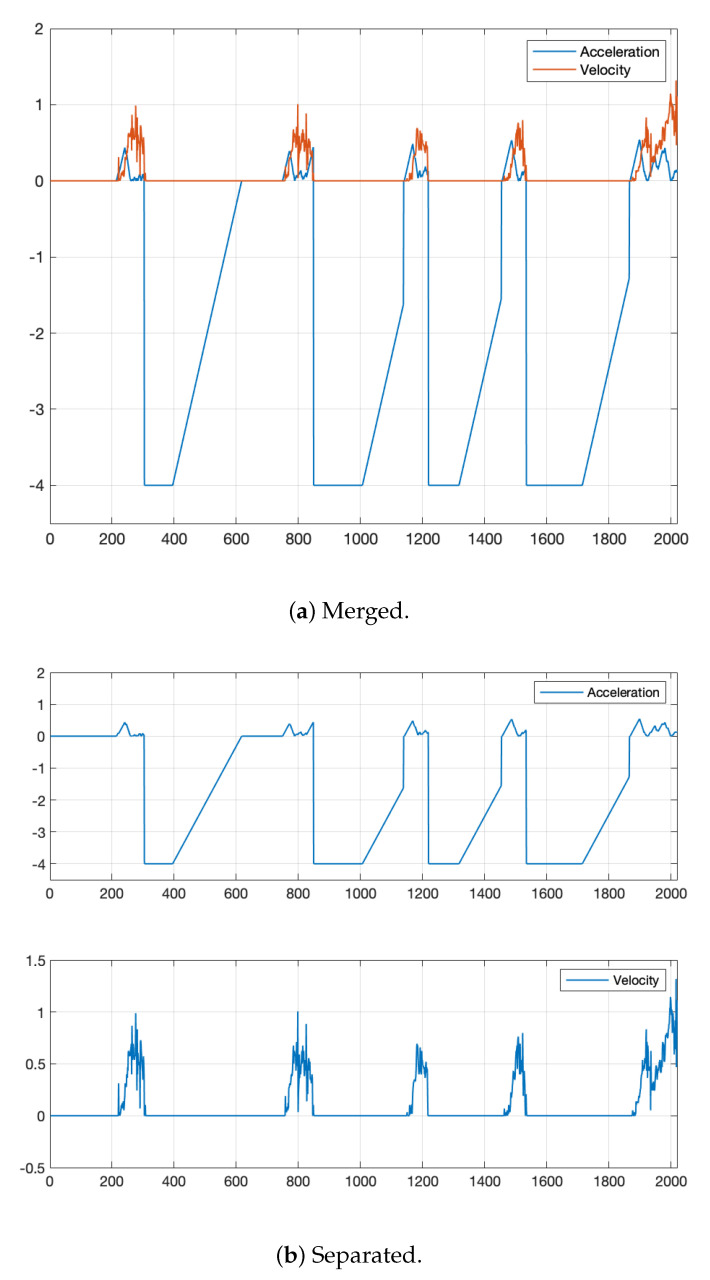
Calculated accelerations and actual velocities.

**Figure 24 sensors-22-09094-f024:**
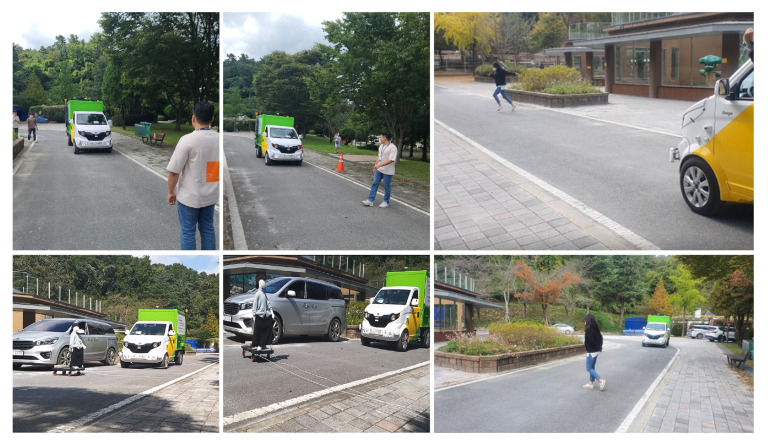
Human recognition test.

**Figure 25 sensors-22-09094-f025:**
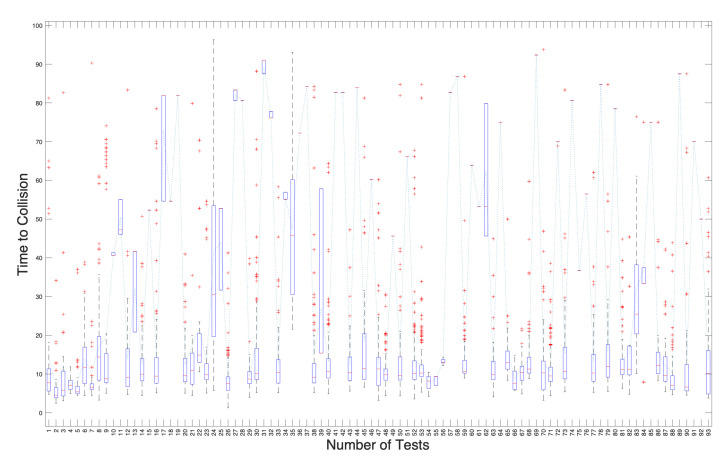
Human recognition result.

**Figure 26 sensors-22-09094-f026:**
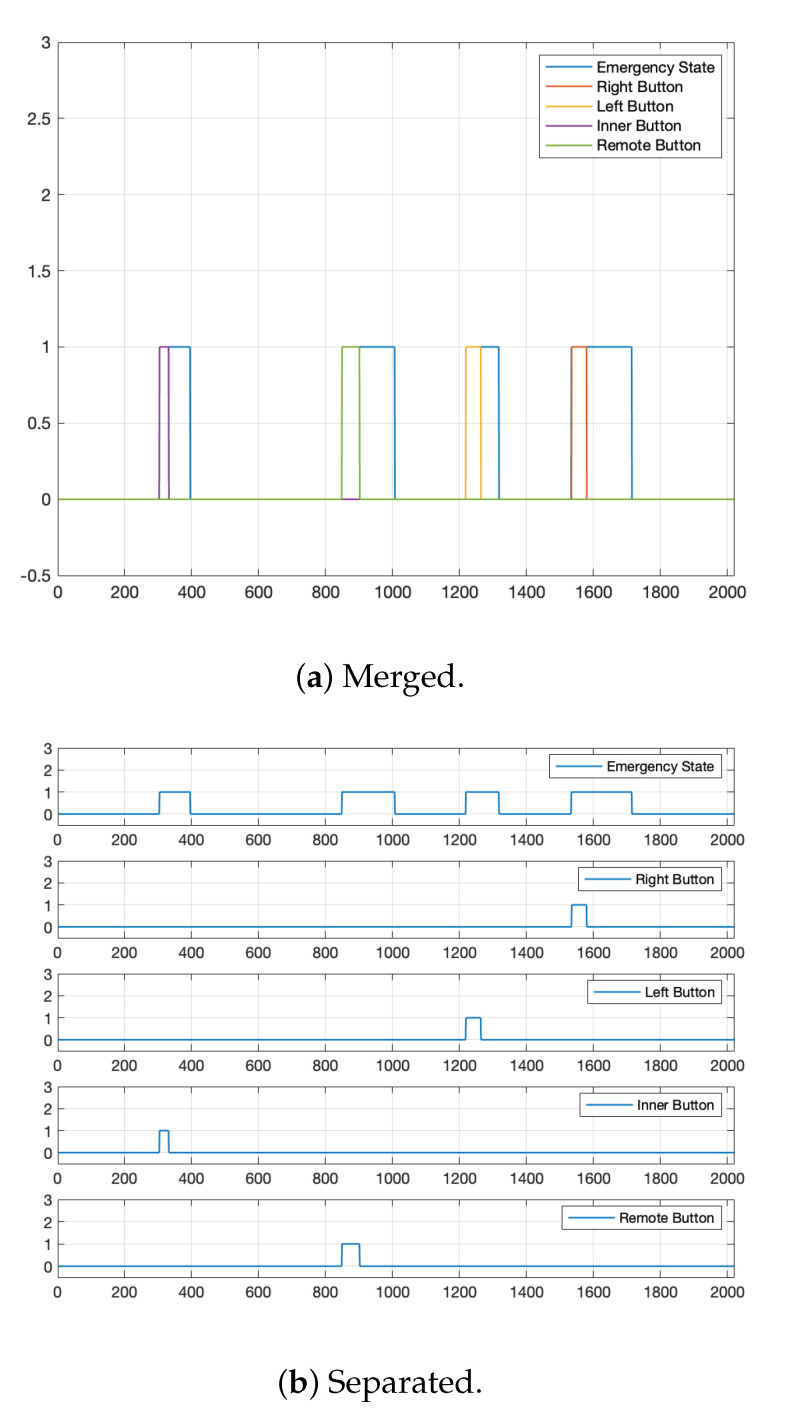
Response of emergency buttons.

**Figure 27 sensors-22-09094-f027:**
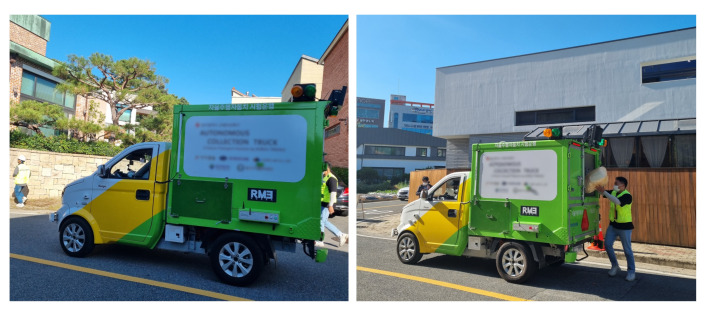
Garbage collection with the autonomous vehicle.

**Table 1 sensors-22-09094-t001:** Garbage collection Test.

Num.	Manual (s)	Automatic (s)
1	993	1156
2	1005	1160
3	1013	1163
4	1011	1161
5	981	1151
6	986	1155
7	1008	1160
8	1002	1159
9	979	1147
10	997	1157
Mean	997.50	1156.90

## Data Availability

Not applicable.

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
