# Peer review of "Development of an Autonomous Driving Vehicle for Garbage Collection in Residential Areas"

_sensors, 2022, doi:10.3390/s22239094_

Round 1

Reviewer 1 Report

In this article authors have proposed a work with title “Development of an autonomous driving vehicle for garbage collection in residential areas”.

Please consider following comments.

1: The abstract lacks the article's main idea and theoretical contribution since it is too generic and vague.

2: The visibility and parameter definitions of results presented as figures should be improved.

3: The comparison of the figures is not given in the right or expected format. When two graphs are compared in figures, the error plot is not displayed. The existence of duplicate results in the form of merged/separated arguments was needless.

4: The authors haven't provided any comparisons to alternative methodologies.

5: In this situation, it appears that the current algorithms are being applied to a separate problem. Therefore, if writers truly believe that their concept is original, they should explain it both in their work and in their response to this comment.

6: For a specific problem or example, it is also important to discuss the cost, system optimization, and computation time.

Author Response

Response to Reviewer 1 Comments

Point 1: The abstract lacks the article's main idea and theoretical contribution since it is too generic and vague.

Response 1: We have revised this paper's abstract more clearly to express our purpose and contribution to this research.

Point 2: The visibility and parameter definitions of results presented as figures should be improved.

Response 2: We resized all figures to be visible clearly and added explanations to avoid confusion about the parameters of the graphs.

Point 3: The comparison of the figures is not given in the right or expected format. When two graphs are compared in figures, the error plot is not displayed. The existence of duplicate results in the form of merged/separated arguments was needless.

Response 3: We excluded the 'Separated' graph from the 'Steering Result' graph. However, the 'Separate' graphs of 'Velocity Result' and 'Emergency Button Result' were maintained. The reason for maintaining the 'Separated' graph of 'Velocity Result' is to avoid confusion because each graph represents a different unit. And the reason for maintaining the 'Separated' graph of 'Emergency Button Result' is that all signals from the 'Merged' graph overlapped, making it difficult to see when the signals for each button come in.

Point 4: The authors haven't provided any comparisons to alternative methodologies.

Response 4: In this paper, we developed an autonomous vehicle for garbage collection in residential areas. Therefore, a proper comparison for this paper requires a comparison with autonomous vehicles similar to those for this purpose. Unfortunately, we could not find any vehicles or research papers allowing this comparison from our perspective. In the case of commercial vehicles capable of autonomous or semi-autonomous driving, information such as vehicle estimation, recognition results, and accurate speed can only be obtained directly with the manufacturer's cooperation. Moreover, some experiments we conducted in this paper cannot be performed with these vehicles for safety reasons. In addition, since this vehicle is under development, and various components such as sensor structure, wiring, and system structure are jointly combined, direct comparison with various studies on autonomous vehicles was not possible. Therefore, in this paper, instead of comparing with other studies, we tried to verify the developed vehicle by conducting various experiments in the target environment.

Point 5: In this situation, it appears that the current algorithms are being applied to a separate problem. Therefore, if writers truly believe that their concept is original, they should explain it both in their work and in their response to this comment.

Response 5: The purpose of the vehicle developed in this paper is to help the developed vehicle smoothly operate in the residential area we are targeting and collect garbage. The minimum conditions for this are that the vehicle must locate itself, recognize objects, adjust speed, and stop in case of danger. Also, efficiency measurements are needed to ensure that the developed vehicles help with garbage collection. All the other experiments performed in this paper are necessary to operate our developed autonomous driving vehicle in the target environment

Point 6: For a specific problem or example, it is also important to discuss the cost, system optimization, and computation time.

Response 6: As for the cost, it is not included because it is confidential and difficult to estimate accurately, and the cycle and variable settings of the entire system are added to the 'Experiments' section.

Reviewer 2 Report

The manuscript is well structured and well argued. However, several rectifications and modifications are required to ensure that its quality stands up to this reputed journal.  

1.      The authors have carried out the experimental research on the development of an autonomous driving vehicle for garbage collection to support the garbage collection task of workers in residential areas, which is the strong point of the manuscript.

2.      The English language must be improved. There are several grammatical errors as one goes through the manuscript that requires rectification. Most of the sentences convey no proper meaning and could be off-putting to the readers and practitioners.

3.      The first section introduces a basic outlook on importance of autonomous driving vehicles and deep learning based research for introducing into the vehicle and a brief literature review of performance of developed autonomous vehicles is presented.

4.      ODD is to be expanded in abstract. The resolution of all the figures can be improved.

5.      Explain the criteria to select the values of kp and ki (Equation 35) in the calculation of steering angle.

6.      How the PID controller used for vehicle speed is tuned? Suggested to include in the manuscript.

7.      To keep things fair, a brief discussion of the demerits with the experimental research should be provided.

8.      The manuscript has the potential to be improved and requires major rectifications. With that being said, I wish the authors all the best on their endeavor to improve the quality of the manuscript.

Author Response

Response to Reviewer 2 Comments

Point 1: The authors have carried out the experimental research on the development of an autonomous driving vehicle for garbage collection to support the garbage collection task of workers in residential areas, which is the strong point of the manuscript.

Response 1: That statement is very accurate. The vehicle in this paper was developed to support garbage collection in residential areas. We modified the 'Introduction' part to clarify this point for readers.

Point 2: The English language must be improved. There are several grammatical errors as one goes through the manuscript that requires rectification. Most of the sentences convey no proper meaning and could be off-putting to the readers and practitioners.

Response 2: We double-checked all English grammar for the entire paper and corrected any mistakes. At the same time, we removed unnecessary parts and added necessary parts to convey the meaning well to the readers.

Point 3: The first section introduces a basic outlook on importance of autonomous driving vehicles and deep learning based research for introducing into the vehicle and a brief literature review of performance of developed autonomous vehicles is presented.

Response 3: In the 'introduction' part, we explained the need for autonomous vehicles and deep learning research related to autonomous vehicles. Then, we mentioned the performance of current autonomous vehicles, Briefly.

Point 4: ODD is to be expanded in abstract. The resolution of all the figures can be improved.

Response 4: We mentioned the part about ODD in the abstract. Also, we adjusted the size of all figures.

Point 5: Explain the criteria to select the values of kp and ki (Equation 35) in the calculation of steering angle.

Response 5: We briefly mentioned in the 'Executor' section how these hyperparameters are determined (line. 735).

Point 6: How the PID controller used for vehicle speed is tuned? Suggested to include in the manuscript.

Response 6: We briefly mentioned in the 'Experiments' section how our system is tuned (line. 811).

Point 7: To keep things fair, a brief discussion of the demerits with the experimental research should be provided.

Response 7: We divided the 'Conclusion and Future Works' part into 'Conclusion' and 'Future Works' and briefly summarized the shortcomings of the vehicle we developed in the 'Conclusion' part.

Point 8: The manuscript has the potential to be improved and requires major rectifications. With that being said, I wish the authors all the best on their endeavor to improve the quality of the manuscript.

Response 8: We actively accepted the review and revised the entire paper to make it more transparent and easier to explain. In addition, we endeavored to correct the grammar and context as much as possible to improve the quality of the entire paper.

Round 2

Reviewer 1 Report

I accept this article for the journal in its most recent form. Questions that were raised about the earlier version of the manuscript have been addressed by the authors. It is expected that the authors will contribute an improvement in the form of research articles that can presently not be included in this article.